# Endomitosis controls tissue-specific gene expression during development

**Lotte M. van Rijnberk, Ramon Barrull-Mascaró, Reinier L. van der Palen, Erik S. Schild, Hendrik C. Korswagen, Matilde Galli***

Hubrecht Institute, Royal Netherlands Academy of Arts and Sciences and University Medical Center Utrecht, Utrecht, the Netherlands

* m.galli@hubrecht.eu

**Data Availability Statement:** All data that was used for the main text and supporting information is available. The RNA-seq data generated in this

## Abstract

Polyploid cells contain more than 2 copies of the genome and are found in many plant and animal tissues. Different types of polyploidy exist, in which the genome is confined to either 1 nucleus (mononucleation) or 2 or more nuclei (multinucleation). Despite the widespread occurrence of polyploidy, the functional significance of different types of polyploidy is largely unknown. Here, we assess the function of multinucleation in *Caenorhabditis elegans* intestinal cells through specific inhibition of binucleation without altering genome ploidy. Through single-worm RNA sequencing, we find that binucleation is important for tissue-specific gene expression, most prominently for genes that show a rapid up-regulation at the transition from larval development to adulthood. Regulated genes include vitellogenins, which encode yolk proteins that facilitate nutrient transport to the germline. We find that reduced expression of vitellogenins in mononucleated intestinal cells leads to progeny with developmental delays and reduced fitness. Together, our results show that binucleation facilitates rapid up-regulation of intestine-specific gene expression during development, independently of genome ploidy, underscoring the importance of spatial genome organization for polyploid cell function.

## Introduction

Polyploidization occurs in many plant and animal cells as part of a developmental program, where it is crucial to increase cell size and metabolic output, as well as to maintain the barrier function of certain tissues. For example, polyploidization of the giant neurons of *Limax* slugs allows them to reach the enormous size they need to transmit signals over large distances, and polyploidization of the *Drosophila* subperineurial glial cells ensures the integrity of the blood–brain barrier during organ growth [1,2]. In vertebrates, polyploid cells are present in many organs, such as the liver, blood, skin, pancreas, placenta, and mammary glands, where they play important functions in tissue homeostasis, regeneration, and in response to damage [3–13]. Interestingly, polyploid cells can either be mononucleated, such as megakaryocytes and trophoblast giant cells [14,15], or they can be multinucleated, such as cardiomyocytes and mammary epithelial cells [8,16], but how genome partitioning in either 1 or multiple nuclei affects polyploid cell function is yet unknown.

study has been deposited at the Gene Expression Omnibus (GSE169330).

**Funding:** Dutch Research Council (Nederlandse Organisatie voor Wetenschappelijk Onderzoek, https://www.nwo.nl/en): NWO-Veni grant to MG (016.Veni.181.016). Human frontiers science program organization (https://www.hfsp.org): HFSP Career Development Award to MG (CDA00018/2017-C) and HFSP Program Grant to HCK and ESS (RGP0030/2016). Netherlands Organisation for Health Research and Development (Nederlandse organisatie voor gezondheidsonderzoek en zorginnovatie, https://www.zonmw.nl/en): ZonMw Enabling Technologies Hotel grant to MG (ETH 40-43500-98-4087). The funders had no role in study design, data collection and analysis, decision to publish, or preparation of the manuscript.

**Competing interests:** The authors have declared that no competing interests exist.

**Abbreviations:** AID, auxin-inducible degron; DIC, differential interference contrast; FPKM, fragments per kilobase of exon per million mapped fragments; GO, Gene Ontology; NGM, nematode growth medium; PI, propidium iodide; smFISH, single molecule fluorescence in situ hybridization.

Somatic polyploidy can arise either by cell fusion or by noncanonical cell cycles in which cells replicate their DNA but do not divide. Two types of noncanonical cell cycles that result in polyploidy have been described: endoreplication and endomitosis [17]. In endoreplication, M phase is skipped, resulting in cycles of DNA replication (S phase) and gap (G) phases without intervening mitosis and cytokinesis. Endoreplicative cell cycles result in large, mononucleated cells. In endomitosis, cells do enter mitosis, but do not undergo cell division, resulting in polyploid cells with either a single nucleus or 2 nuclei, depending on whether M phase is aborted before or after initiation of sister chromosome segregation (which normally occurs during anaphase) [7,11,18]. The existence of multiple types of polyploid cells (e.g., mononucleated or binucleated) suggests that there may be a functional difference between these different types of polyploidy. However, testing the functional significance of multinucleation has been challenging due to the complexity of many polyploid tissues and a lack of tools to specifically alter noncanonical cells cycles without affecting any other cells or tissues.

The *Caenorhabditis elegans* intestine provides an ideal model system to study distinct types of polyploidy, as intestinal cells undergo both endomitosis and endoreplication cycles in a highly tractable manner at defined moments during larval development (Fig 1A and 1B) [19]. Such consecutive cycles of endomitosis and endoreplication give rise to large, binucleated polyploid cells that make up the adult intestine. Here, we develop a method to inhibit intestinal endomitosis using auxin-inducible degradation of key mitotic regulators, allowing us to study the function of binucleation at both the cellular and tissue level. We find that animals with mononucleated instead of binucleated intestinal cells have decreased fitness due to defects in the expression of a group of tissue-specific genes, including the vitellogenin genes. Vitellogenin genes are normally rapidly expressed during the maturation of intestinal cells at the end of larval development, indicating that rapid up-regulation of tissue-specific genes requires binucleation of polyploid cells. Importantly, this rapid up-regulation of intestine-specific genes is important to support progeny development. Together, our results show that binucleation is important for correct functioning of a polyploid tissue and that partitioning of genomes into multiple nuclei allows efficient and rapid up-regulation of gene expression during development.

## Results

### Auxin-inducible degradation of mitotic regulators prevents binucleation

To investigate the function of binucleation, we developed a system in which we can specifically perturb intestinal endomitosis, without affecting other cell cycles within developing *C. elegans* larvae. We employed 2 mechanistically distinct approaches to block binucleation of the intestine; we either depleted CDK-1, which is essential for mitotic entry, or KNL-1, a conserved kinetochore protein that is required for chromosome segregation during M phase. CDK-1 inhibition is known to be a key mechanism to initiate endoreplication, and degrading CDK-1 essentially converts the endomitosis cycle to an endoreplication cycle, preventing mitotic entry and subsequent binucleation [14,17,20,21]. In contrast, KNL-1 inhibition does not affect mitotic entry but prevents chromosome segregation and partitioning of sister chromatids into 2 nuclei [22,23]. To deplete CDK-1 and KNL-1 specifically in intestinal cells and only during the time when endomitosis occurs, we made use of the auxin-inducible degradation system. In this system, proteins tagged with an auxin-inducible degron (AID) are degraded only in the presence of auxin and the F Box protein TIR1 [24–26]. By using a strain that expresses TIR1 under control of the intestine-specific *Pges-1* promoter and exposing animals to auxin only during the time that endomitosis occurs, we can specifically inhibit intestinal binucleation without affecting canonical cell cycles of intestinal or other cells.

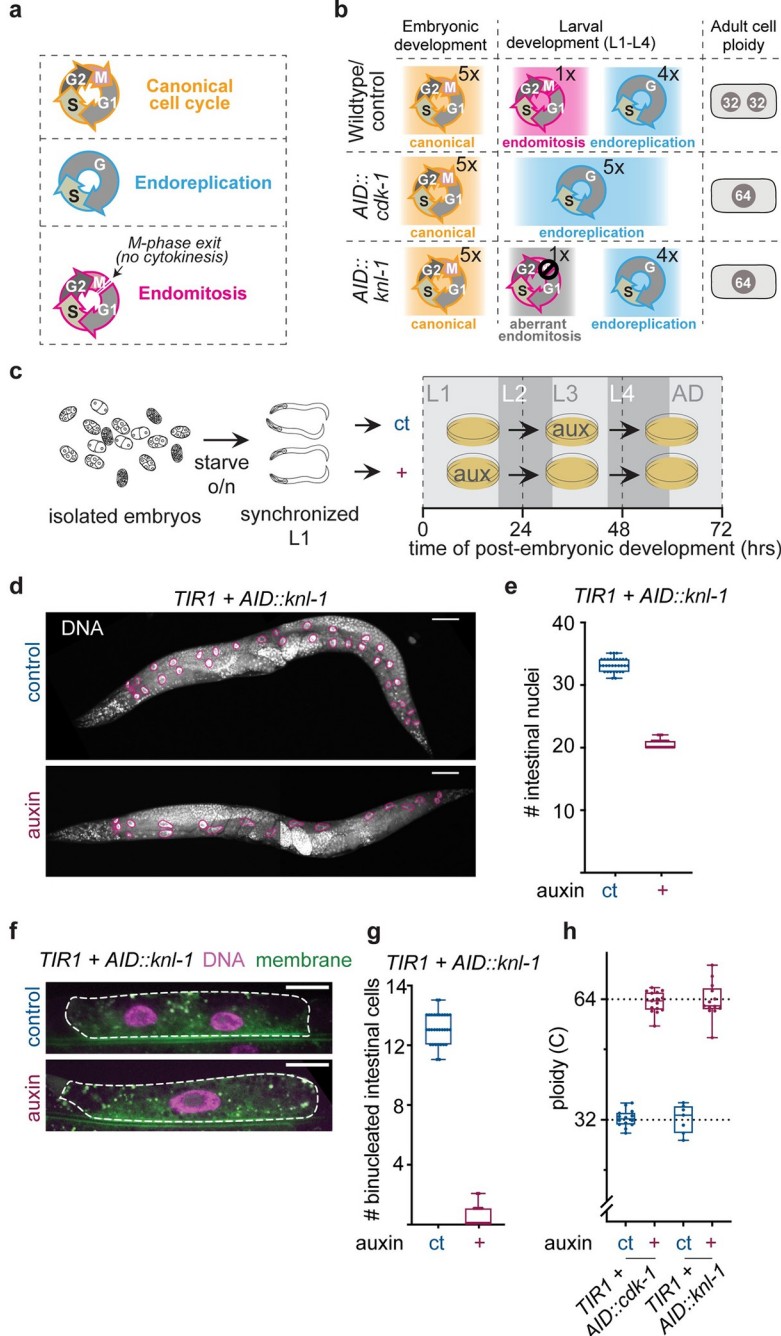

**Fig 1. Inducible and tissue-specific degradation of mitotic proteins specifically prevents binucleation of the *C. elegans* intestine. (A)** Three types of cell cycles that take place during intestinal development in *C. elegans*. **(B)** Overview of intestinal cell cycles in wild-type/control animals and upon intestinal depletion of CDK-1 or KNL-1. In wild-type *C. elegans*, 20 intestinal cells are formed during embryogenesis by canonical cell cycles. During the first larval stage (L1) intestinal cells undergo one round of endomitosis creating binucleated cells. Thereafter, intestinal cells undergo one round of endoreplication at the end of each larval stage, leading to adult intestinal cells with two 32C nuclei (and thus a cellular ploidy of 64C DNA content). Upon inhibition of either CDK-1 or KNL-1 during endomitosis using the auxin-inducible degradation system, endomitotic binucleation is blocked, and a mononucleated polyploid cell is generated. Importantly, cellular ploidy remains the same as wild-type conditions (64C). **(C)** Schematic overview of experimental procedure. Mixed stage embryos are isolated from adult hermaphrodites containing intestinally expressed TIR1 (*Pges-1::TIR1*) and either *AID::cdk-1* or *AID::knl-1* and starved overnight to yield a synchronized population of arrested L1 animals. The population of starved L1 animals is split into 2 conditions: an auxin (+) condition and a control (ct) condition. For the auxin condition, animals are grown on plates containing

auxin for the first 24 hours of postembryonic development, when endomitosis normally occurs, and transferred to plates without auxin after this period. Animals in control conditions are grown on plates without auxin for the first 24 hours of development and transferred to plates containing auxin for 24 to 48 hours of postembryonic development, when intestinal endomitosis has already occurred and neither KNL-1 or CDK-1 are required in the intestine. **(D)** DAPI stainings of adult hermaphrodites containing *Pges-1::TIR1; AID::knl-1* and grown under auxin or auxin-control conditions. Intestinal nuclei are outlined in magenta. Scale bar is 50 μm. **(E)** Number of intestinal nuclei in adult hermaphrodites containing *Pges-1::TIR1; AID::knl-1* and grown under auxin (+) or auxin-control (ct) conditions. Boxplots indicate the median and 25th to 75th percentile, error bars indicate min to max values, and individual values are shown as dots. **(F)** Fluorescent images of intestinal H2B-mCherry (DNA) and GFP-PH (membrane) in intestinal ring 3 of *Pges-1::TIR1; AID::knl-1* animals grown on auxin or auxin-control conditions. Dashed line indicates cell outline. Scale bar is 20 μm. **(G)** Number of binucleated intestinal cells in adult *Pges-1::TIR1; AID::knl-1* hermaphrodites grown on auxin (+) or auxin-control (ct) conditions. Boxplots indicate the median and 25th to 75th percentile, error bars indicate min to max values, and individual values are shown as dots. **(H)** Quantification of intestinal nuclear ploidy in binucleated control (ct, *n* = 49 for *Pges-1::TIR1; AID::cdk-1* strain and *n* = 28 for *Pges-1::TIR1; AID::knl-1* strain) or mononucleated intestinal cells (+ auxin, *n* = 29 for *Pges-1::TIR1; AID::cdk-1* strain and *n* = 26 cells for *Pges-1::TIR1; AID::knl-1* strain). Ploidy is measured by total fluorescence intensity of propidium iodide (PI) DNA staining in intestinal ring 3 nuclei (Int3D and Int3V) and normalized to proximal 2C nuclei. Each dot represents the average ploidy of individual Int3D/V nuclei in one animal. Boxplots indicate the median and 25th to 75th percentile, error bars indicate min to max values, and individual values are shown as dots. Underlying data can be found in S1 Data. AID, auxin-inducible degron.

We generated AID knock-ins on the *cdk-1* and *knl-1* genes using CRISPR-mediated gene targeting and tested whether auxin induced depletion of CDK-1 or KNL-1 during endomitosis was able to block binucleation. In wild-type animals, endomitosis takes place at the end of the first larval stage in 12 to 14 of the 20 intestinal cells, resulting in 20 intestinal cells with 32 to 34 nuclei [19]. To control for nonspecific effects of auxin on development, we included an auxin control in each experiment, in which animals were treated with auxin for the same duration, but during a time in development in which CDK-1 or KNL-1 are not required in the intestine (Fig 1B and 1C). To assess the effect of auxin treatment on intestinal binucleation, we analyzed animals grown with auxin during endomitosis (hereafter referred to as auxin-treated animals or "+") or in the auxin-control condition (hereafter referred to as auxin-control animals or "ct") and found that auxin-treated animals contain 20 to 22 intestinal nuclei, in contrast to the 32 to 34 nuclei that are present in auxin-control animals (Fig 1D and 1E). Of the 12 to 14 intestinal cells that are known to undergo endomitosis and become binucleated in wild-type intestines, we found on average 13 cells to be binucleated in auxin-control animals compared to 0 binucleated cells in auxin-treated animals (Fig 1F and 1G). We next performed a quantitative DNA staining using PI and found that auxin-treated *AID::cdk-1* and *AID::knl-1* animals have a 64C DNA content, which is exactly double the nuclear ploidy of controls (Fig 1H). Thus, by inhibiting CDK-1 and KNL-1 during endomitosis, we can block binucleation and generate animals with mononucleated intestinal cells with the same cellular ploidy as wild-type binucleated intestinal cells.

Depletion of KNL-1 is known to prevent chromosome segregation in anaphase by blocking formation of kinetochore microtubule attachments [22]. To exclude that KNL-1 knockdown results in formation of micronuclei caused by chromosome missegregations, which can lead to DNA damage, cellular stress, and potentially arrest cells in the following cell cycle [27–30], we performed live-cell imaging of fluorescently labeled chromosomes in KNL-1 depleted cells. In all cells, KNL-1 depletion prevented chromosome segregation and resulted in mononucleation, but did not extend the duration of mitosis or result in the formation of micronuclei (S1A and S1B Fig, S1 and S2 Movies). After endomitosis, cells go through multiple rounds of endoreplication in which newly formed replicated chromosomes remain clustered together. Wild-type intestinal cells that have undergone endomitosis contain 2 nuclei, each containing 2 sets of chromosome clusters, which can be visualized by chromosomal LacO/LacI tagging (S1C Fig). As expected, we observed on average 4 chromosome clusters in nuclei of mononucleated

KNL-1 depleted cells (S1D Fig), indicating that all copies of the labeled chromosome were present in a single nucleus. Finally, using a fluorescent cell cycle marker (*Pges-1*:: *CYB-1^{DB}*:: *mCherry)*, we found that preventing binucleation did not affect the timing of the subsequent endoreplicative S phase (S1E Fig). Taken together, our results indicate that KNL-1 depletion prevents binucleation without inducing detectable chromosomal or cell cycle aberrancies. Thus, this system provides a unique opportunity to study the function of binucleation, without altering the ploidy or number of cells in the tissue of interest.

## Intestinal mononucleation decreases the nuclear surface-to-volume ratio, but does not affect cell size or morphology

To investigate whether mononucleation influences intestinal cell size or morphology, we used fluorescent markers in the *AID*::*knl-1* strain to visualize the cell membranes and intestinal lumen. We found no effect of intestinal mononucleation on animal size, cell size, or lumen morphology (Fig 2A–2H). Because polyploidization has also been shown to influence nuclear morphology [31], we investigated the effect of binucleation on nuclear geometry using a fluorescently labeled nuclear pore protein (NPP-9::mCherry) as a marker for the nuclear membrane (Fig 2I and 2J). We measured nuclear size and found that the nuclear volume was increased with a factor of approximately 2.5 in mononucleated cells, indicating that per cell, the total nuclear volume has more than doubled (Fig 2K). The nuclear surface area was also increased in mononucleated cells, but to a lesser extent, with a factor of 2 (Fig 2L). Consequently, the surface-to-volume ratio decreased, indicating that less surface area is available per volume in mononucleated cells (Fig 2M). To test whether this effect is compensated to any extent by shape alterations or membrane invaginations that enlarge nuclear surface area, such has been observed in other polyploid cells [31], we also measured the ratio between the circumference and area of nuclear sections, but observed a similar effect (S2A–S2D Fig), indicating that there are no large invaginations that compensate for the amount of available nuclear surface area per volume. Thus, although mononucleation does not affect intestinal cell size or morphology, it produces nuclei that are more than twice as big and have an altered surface-to-volume ratio, which could potentially influence nuclear functions.

## Binucleation is important for adult intestinal transcription

To investigate possible transcriptional differences between young adults with mononucleated or binucleated intestines, we performed single-worm RNA sequencing of *AID*::*knl-1* and *AID*:: *cdk-1* auxin-treated and auxin-control animals. Since the intestine is one of the most transcriptionally active and largest tissues in the worm, making up roughly one-third of the animals volume [32], we anticipated that changes in transcription in the intestine would be detectable in whole worm sequencing. We performed differential gene expression analysis for 68 (*AID*::*knl-1*) and 75 (*AID*::*cdk-1*) young adult (72 hours) auxin-treated or auxin-control animals (see Methods for details). For the *AID*::*knl-1* animals alone, 16 percent of genes included in the analysis showed significant and substantial differential expression between worms with a mononucleated or binucleated intestine (420/2,638 genes, Fig 3A, S3A Fig). Sequencing of *AID*::*cdk-1* animals revealed fewer differentially expressed genes between animals with mononucleated or binucleated intestines, which is likely due to a lower complexity of these samples (S3B–S3E Fig). Although it is unclear why the *AID*::*cdk-1* samples had lower complexities, we focused on the overlap in differential gene expression between the *AID*::*knl-1* and *AID*::*cdk-1* strains to identify genes that are differentially expressed in animals with mononucleated intestines (S3F Fig). We found a strong enrichment (*P* = 2e-7) in the overlap between genes significantly and substantially down-regulated in both strains, consisting of 15 genes

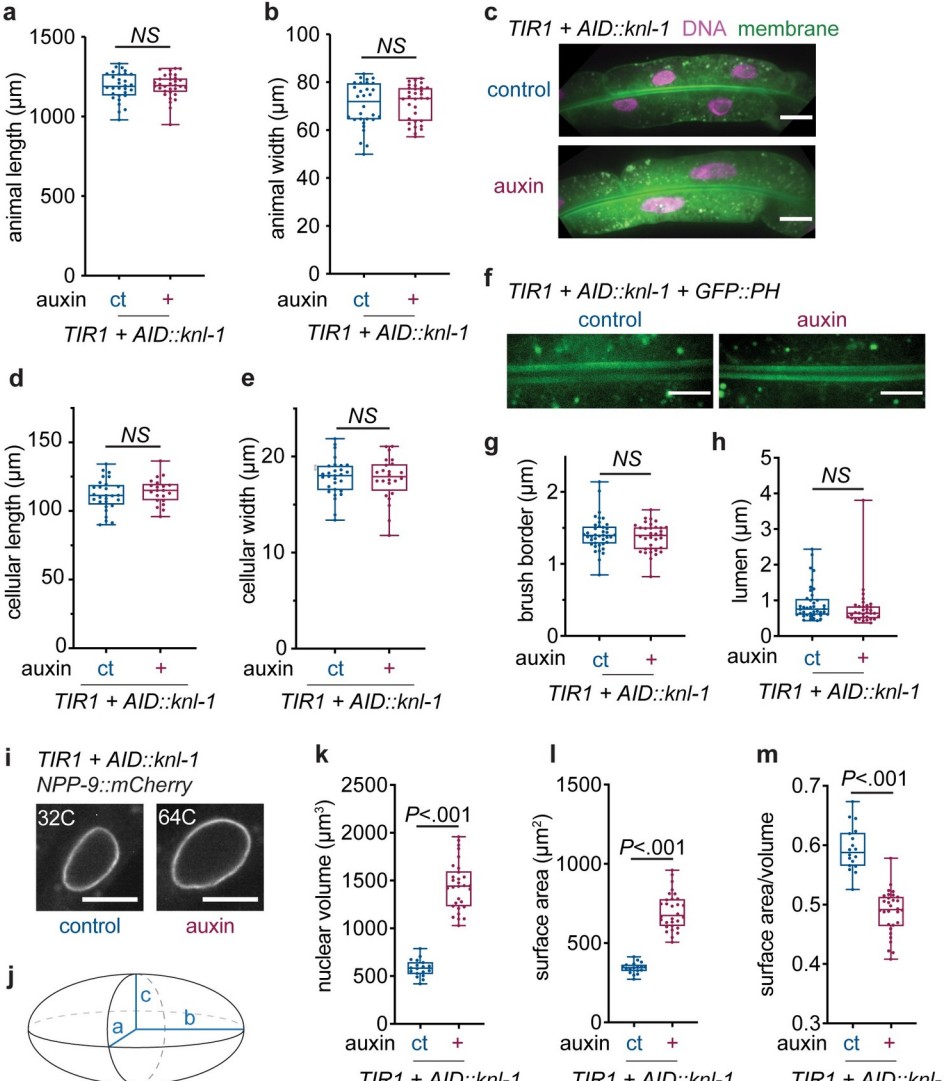

**Fig 2. Intestinal cell mononucleation does not influence worm size, cell size, or intestinal morphology, but generates nuclei that are more than twice as big. (A, B)** Total body length (A) and width (B) of *Pges-1*::*TIR1; AID*::*knl-1* young adults grown under control conditions (ct, *n* = 30) or in the presence of auxin (+, *n* = 32). Boxplots indicate the median and 25th to 75th percentile, error bars indicate min to max values, and individual values are shown as dots. NS, not significant (*P* > 0.05, Student *t* test). **(C)** Representative Z-stack projections of intestinal H2B-mCherry (DNA) and GFP-PH (membrane) in intestinal ring 3 of *Pges-1*::*TIR1; AID*::*knl-1* animals grown on auxin or control conditions, used for quantifications of cell length and width. Scale bar is 20 μm. **(D, E)** Cellular length (D) and width (E) of Int3D/V cells of *Pges-1*::*TIR1; AID*::*knl-1* young adults grown under control conditions (ct, *n* = 58) or in the presence of auxin (+, *n* = 24). Boxplots indicate the median and 25th to 75th percentile, error bars indicate min to max values, and individual values are shown as dots. NS, not significant (*P* > 0.05, Student *t* test). **(F)** Representative Z-stack projections of intestinal GFP-PH (membrane) in the apical membrane of intestinal ring 3 of *Pges-1*::*TIR1; AID*::*knl-1* animals that were grown under control conditions or in the presence of auxin, used for quantifications of brush border and luminal width. Scale bar is 10 μm. **(G, H)** Intestinal brush border (G) and lumen (H) width in intestinal ring 3 of young adult worms grown under control (ct, *n* = 40) or auxin (+, *n* = 34) conditions. Boxplots indicate the median and 25th to 75th percentile, error bars indicate min to max values, and individual values are shown as dots. NS, not significant (*P* > 0.05, Mann–Whitney test). **(I)** Fluorescent images of intestinal NPP-9::mCherry localization in the nuclear membranes of Int3 cells of *Pges-1*::*TIR1; AID*::*knl-1* adults grown under control or auxin conditions, corresponding to a 32C and 64C DNA content, respectively. Scale bar represents 10 μm. **(J)** Schematic depicting an ellipsoid shape and radii a, b, and c used to calculate nuclear volume and surface area. **(K–M)** Boxplots depicting the average nuclear volume (K), surface area (L) and surface-area-to-volume ratio (M) in binucleated (ct, *n* = 60 cells, 18 animals) or mononucleated (+, *n* = 56 cells, 30 animals) cells of *Pges-1*::*TIR1; AID*::*knl-1* animals. Boxplots indicate the median and 25th to 75th percentile, error bars indicate min to max values, and individual values are shown as dots. *P* values were calculated by Mann–Whitney (K, L) or unpaired Student *t* test (M). Underlying data can be found in S1 Data. AID, auxin-inducible degron.

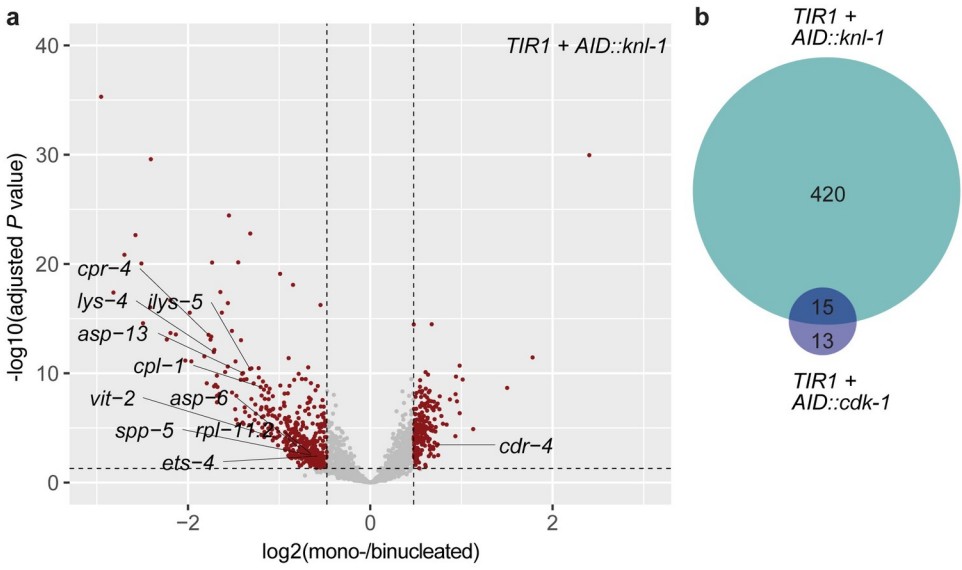

c

| Sequence name | Gene name | Description | Phenotype |
|---|---|---|---|
| C42D8.2 | *vit-2* | Vitellogenin | Several changes in progeny growth |
| F07D10.1 | *rpl-11.2* | Ribosomal protein | Larval arrest/lethal and maternal sterile |
| F08F1.3 | | Membrane protein | |
| F21F8.7 | *asp-6* | Cathepsin E | |
| F22A3.1 | *ets-4* | Transcription factor | Lethal/sterile |
| F22A3.6 | *ilys-5* | Lysozyme | Embryonic lethality |
| F22H10.6 | | | |
| F25A2.1 | | Hydrolase | |
| F28A12.4 | *asp-13* | Cathepsin E | |
| F44C4.3 | *cpr-4* | Cathepsin B | |
| F58B3.1 | *lys-4* | Lysozyme | |
| T03E6.7 | *cpl-1* | Cathepsin L | Embryonic lethality |
| T08A9.9 | *spp-5* | Saposin | Brood size reduction, fat content reduction, larval arrest |
| ZK1320.3 | | | |
| ZK593.3 | | Membrane protein | Fat content reduction |

**Fig 3. Blocking binucleation causes transcriptional down-regulation of intestinally expressed genes in young adult *C. elegans*. (A)** Volcano plot of RNA sequencing data depicting the transcriptional gene up- and down-regulation in worms with a mononucleated intestine, compared to worms with a binucleated (wild type) intestine in *Pges-1::TIR1; AID::knl-1 animals* (*n* = 68). After filtering for coverage, batch consistency, and intestinal expression, 2,638 genes were analyzed for differential gene expression using DEseq. Red dots represent genes significantly differentially expressed (adjusted *P* value < 0.05, top 25% absolute log2(foldchange)). Genes significantly differentially expressed in both *Pges-1::TIR1; AID::knl-1* and *Pges-1::TIR1; AID::cdk-1* comparisons were individually annotated with their gene name (excluding genes without a gene name). **(B)** Venn diagram of the overlap of significantly down-regulated genes in *Pges-1::TIR1; AID::knl-1* and *Pges-1::TIR1; AID::cdk-1* animals. **(C)** Overview of genes that are found to be significantly down-regulated in worms in which intestinal binucleation is blocked, both by degradation of CDK-1 and degradation of KNL-1, including a short description and previously established associated phenotypes as described on WormBase [38]. Underlying data are available at the Gene Expression Omnibus, identifier GSE169330, and in S1 Data. AID, auxin-inducible degron.

(Fig 3B and 3C). One of the genes that stood out was *vit-2*, 1 of the 6 *C. elegans* vitellogenin genes whose levels have been shown to correlate strongly with the growth and fitness traits of *C. elegans* progeny [33]. Vitellogenins are highly expressed in adult intestines and are essential

to mobilize lipids in the intestine for transport to the developing oocytes in the germline, where they contribute to progeny development [34–37]. Because vitellogenins have previously been shown to function redundantly, and down-regulation of individual *vit* genes often leads to up-regulation of others [34], we analyzed the expression of all 6 *vit* genes in our dataset and found that all of them were down-regulated in worms with a mononucleated intestine (S4A– S4F Fig). Moreover, the sum of all *vit* gene expression values showed a stronger decrease than any gene alone (S4G Fig), suggesting that in mononucleated animals, the down-regulation of *vit-2* is not compensated by the transcriptional up-regulation of other vitellogenins.

## Binucleation of the intestine promotes vitellogenin expression and lipid loading into oocytes

To confirm that *vit-2* expression is down-regulated in animals with mononucleated intestinal cells, we generated a *Pvit-2*::<sup>NLS</sup>GFP transcriptional reporter to measure *vit-2* promoter activity at different moments of development. Since vitellogenin expression is absent during larval development and drastically up-regulated at the L4-to-adult transition [34], expression levels are still relatively low at 48 hours of development, around the end of the L4 stage (Fig 4A). Upon adulthood, *vit-2* expression levels increase considerably in both auxin-treated and auxin-control animals. However, *vit-2* promoter activity shows a significant reduction in auxin-treated animals at 72 hours of development. This difference in *vit-2* expression levels is no longer present in older adults, when *vit-2* expression is peaking and becomes similar between animals with mononucleated and binucleated intestinal cells (Fig 4A, 96 and 120 hour time points). To further investigate whether mononucleated intestinal cells have reduced ability to express *vit-2* during early stages of adulthood, we performed single molecule fluorescence in situ hybridization (smFISH) at 54 hours of development, when *vit-2* starts becoming expressed in intestinal cells and it is possible to count single *vit-2* mRNA molecules in intestinal cells. Our analyses revealed that animals with mononucleated intestinal cells had on average fewer *vit-2* mRNAs per cell and reduced nascent transcription in their nuclei (Fig 4B–4D). Consistent with a transcriptional down-regulation of *vit-2* in animals with mononucleated intestinal cells, we also found lower levels of VIT-2 protein in embryos derived from mothers with mononucleated intestines (Fig 4E). Again, VIT-2 levels were significantly lower in embryos derived from auxin-treated 72-hour old adults and, similar to the *vit-2* promoter activity, the difference between auxin-treated and control animals was no longer present on subsequent days of adult development.

Because vitellogenins are required to transport lipids from the intestine to the germline, we investigated whether decreased vitellogenin levels resulted in a reduction of lipid loading into oocytes. For this, we used BODIPY staining to quantify lipid levels in embryos and young adult worms. We observed lower lipid levels in embryos from auxin-treated worms, consistent with reduced lipid loading into oocytes of animals with mononucleated intestines (Fig 4F). Moreover, we found higher amounts of lipids in the intestines of adults with mononucleated intestines (Fig 4G), indicating that specifically the transport of lipids is impaired, rather than lipid production or uptake. The effect of mononucleation on lipid levels was more striking than the decrease that we observed in *Pvit-2*::<sup>NLS</sup>GFP expression, which is consistent with the notion that vitellogenins function in 2 distinct yolk complexes, and that multiple vitellogenins are down-regulated in mononucleated animals. Together, our findings indicate that expression of vitellogenins is hampered by blocking binucleation in the intestine, resulting in reduced lipid loading into developing oocytes.

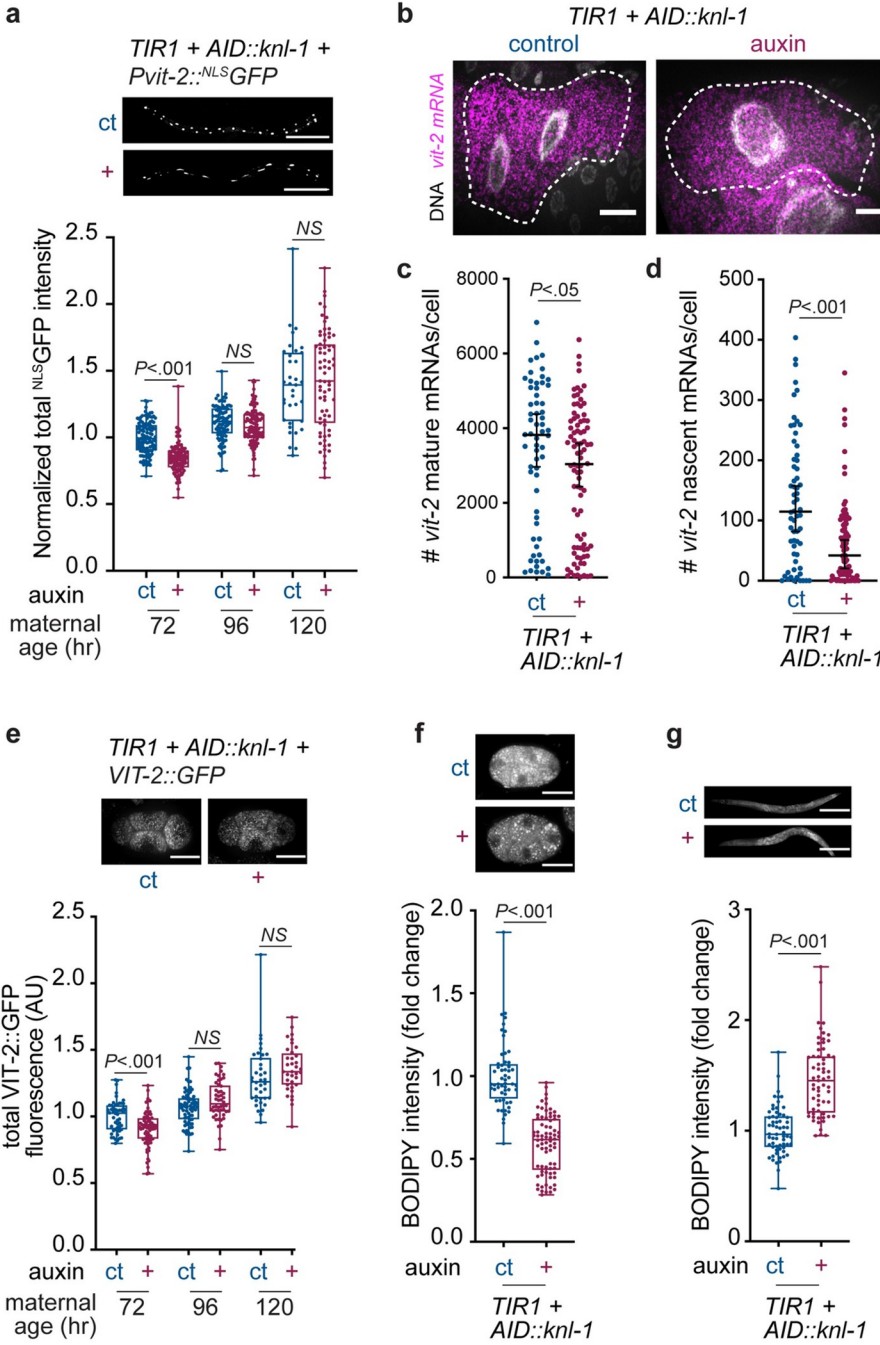

**Fig 4. Perturbation of binucleation decreases yolk protein expression and lipid transport to the germline. (A)**
Fluorescence images and normalized total fluorescence intensities of a *Pvit-2::$^{NLS}$GFP* transcriptional reporter at
different moments during adult development for *Pges-1::TIR1; AID::knl-1* animals grown under control (ct, *n* = 53 to
111) or auxin (+, *n* = 71 to 111) conditions, in 5 replicate experiments. Each data point represents the normalized
fluorescence intensity of one worm. Boxplots indicate the median and 25th to 75th percentile, error bars indicate min
to max values, and individual values are shown as dots. *P* values were calculated by Student *t* test. NS, not significant
(*P* > 0.05). **(B–D)** smFISH analyses of *vit-2* mRNAs in *Pges-1::TIR1; AID::knl-1* animals at 54 hours of development,
grown under control (ct) or auxin (+) conditions. (B) Z-stack projections of *vit-2* smFISH (magenta) in Int3D/V cells.
Animals contain an intestinal membrane marker (*Pges-1::GFP-PH*) that was used to determine cell boundaries (dashed
line). Scale bar represents 10 μm. (C) Quantification of number of cellular *vit-2* mRNAs in Int3D/V cells of animals
grown under control (ct, *n* = 62) or auxin (+, *n* = 88) conditions in 2 replicate experiments. Scatter plot with median
(middle line) and 95% confidence interval (error bars). Each dot represents the number of *vit-2* mRNAs in a single cell.
*P* value was calculated by Mann–Whitney test. (D) Quantification of number of nascent *vit-2* transcripts per cell in

*Pges-1*::*TIR1; AID*::*knl-1* animals grown under control (ct) or auxin (+) conditions. Scatter plot with median (middle line) and 95% confidence interval (error bars). Each dot represents the number of nascent *vit-2* mRNAs in a single cell. *P* value was calculated by Mann–Whitney test. (**E**) Fluorescence images and boxplots showing total endogenous VIT-2::GFP fluorescence intensity in early embryos (1-cell stage to 4-cell stage) derived from control (ct, *n* = 38 to 71) or auxin (+, *n* = 36 to 71) *Pges-1*::*TIR1; AID*::*knl-1* animals, in 2 replicate experiments. Boxplots indicate the median and 25th to 75th percentile, error bars indicate min to max values, and individual values are shown as dots. *P* values were calculated by Mann–Whitney test. NS, not significant (*P* > 0.05). (**F**) Z-stack projection of fluorescence images and boxplots of normalized total fluorescence intensities of BODIPY lipid staining of early embryos isolated from young adult (72 hours) *Pges-1*::*TIR1; AID*::*knl-1* animals grown under control (ct, *n* = 56) or auxin (+, *n* = 77) conditions in 3 replicate experiments. Boxplots indicate the median and 25th to 75th percentile, error bars indicate min to max values, and individual values are shown as dots. *P* values were calculated by Mann–Whitney test. (**G**) Z-stack projection images and boxplots of normalized fluorescence intensities of BODIPY lipid staining of *Pges-1*::*TIR1; AID*::*knl-1* young adults (72 hours) that were grown under control (ct, *n* = 62) or auxin (+, *n* = 63) conditions in 2 replicate experiments. Boxplots indicate the median and 25th to 75th percentile, error bars indicate min to max values, and individual values are shown as dots. *P* values were calculated by Mann–Whitney test. Underlying data can be found in S1 Data. AID, auxin-inducible degron; smFISH, single molecule fluorescence in situ hybridisation.

## Intestinal binucleation enhances *C. elegans* fitness

Abundant expression of vitellogenins in the adult maternal *C. elegans* intestine is important to support postembryonic development and fertility of the offspring [33,36]. To assess whether intestinal binucleation is functionally important to generate sufficient vitellogenins to promote progeny fitness, we performed a competitive fitness assay with animals that have either mononucleated or binucleated intestinal cells. To this end, we generated strains with either a *Pmyo-2*::*mCherry* or a *Pmyo-2*::*GFP* pharyngeal marker in addition to the *AID*::*knl-1* or *AID*::*cdk-1* alleles. This allowed us to follow the progeny of animals with mononucleated or binucleated cells over several generations. By mixing equal amounts of animals with mononucleated (auxin-treated, +) or binucleated intestines (control condition, ct) on plates and counting the proportion of GFP and mCherry positive progeny after several generations, we could determine how binucleation influences reproductive fitness (Fig 5A). It is important to note that auxin was only added during development of the parental worms and that progeny were not treated with auxin and their intestinal cells were thus binucleated. To control for fitness differences due to the *Pmyo-2*::*mCherry* or *Pmyo-2*::*GFP* markers, we included experiments with reversed fluorescent markers and used measurements of the fitness difference between *Pmyo-2*::*mCherry* and *Pmyo-2*::*GFP* control worms to normalize our data (see Methods for details). Using the *AID*::*knl-1* strain, we found that after one generation, animals with a wild-type intestine had given rise to an average of 57.0% (±14.4%) of the population, while animals with a mononucleated intestine only gave rise to 43.0% (±14.4%) of the population (Fig 5B). Similar numbers were obtained with the *AID*::*cdk-1* strain (Fig 5C). These results demonstrate that animals with binucleated intestines have a significant fitness advantage over animals with mononucleated intestines.

To understand how perturbation of binucleation affects fitness, we investigated several aspects of worm growth and reproduction upon inhibition of binucleation. When analyzing *AID*::*knl-1* and *AID*::*cdk-1* animals, we noticed that the *AID*::*cdk-1* strain had smaller brood sizes and increased embryonic lethality compared to wild-type animals, both in the presence or absence of auxin, suggesting that the AID tag compromises CDK-1 function resulting in a weak hypomorph (S5A–S5C Fig). Nonetheless, when comparing animals with mononucleated or binucleated intestines in either the *AID*::*cdk-1* or *AID*::*knl-1* background, we found no reduction in brood sizes in animals with mononucleated intestines (Fig 5D, S5C Fig). However, we did find a significant delay in progeny growth when examining eggs that were laid between 72 and 96 hours of development, corresponding to the first and second day of adulthood (Fig 5E). This delay was not due to growing animals on auxin or the presence of TIR1, as

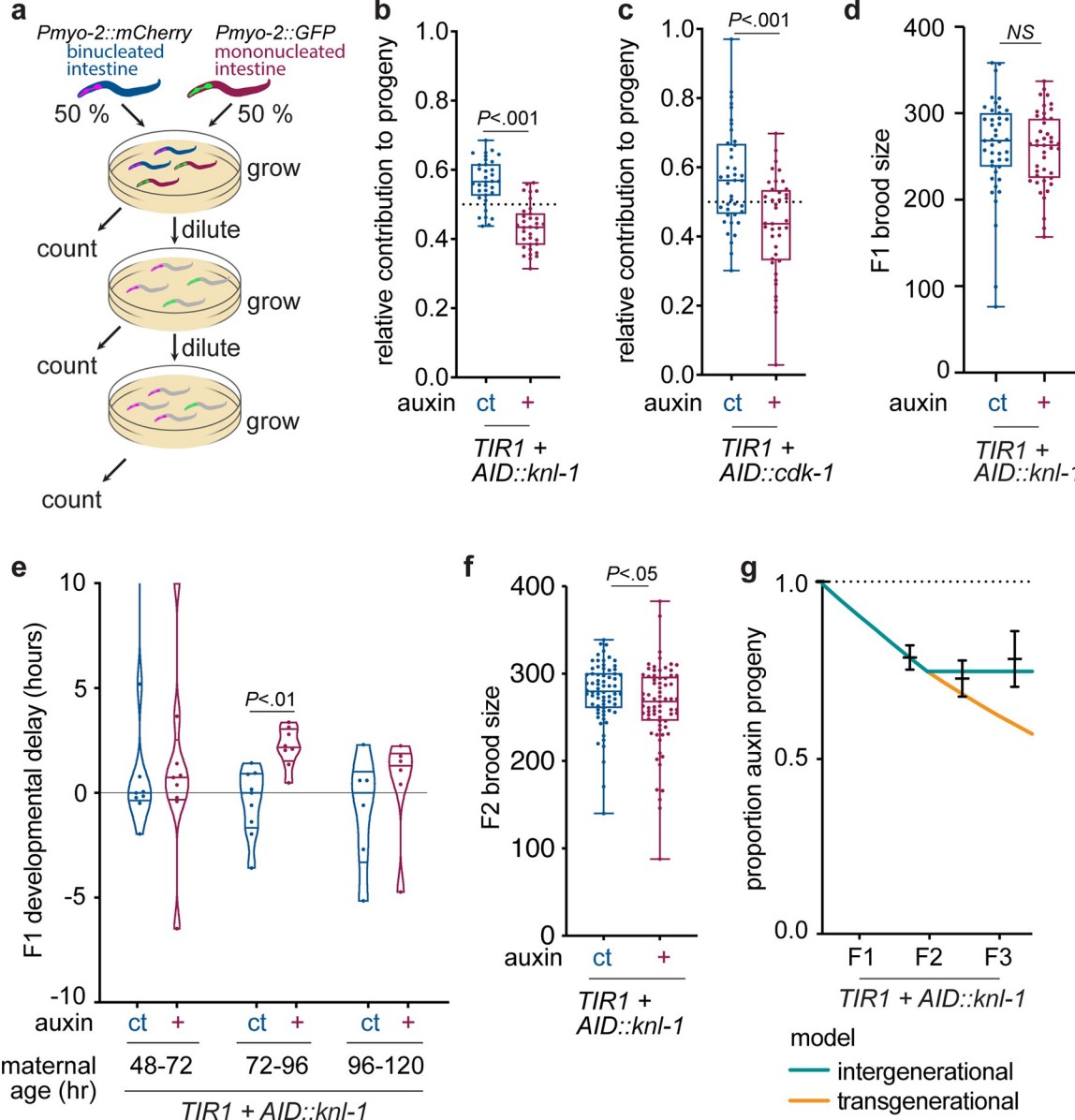

**Fig 5. Inhibition of intestinal binucleation reduces relative reproductive fitness. (A)** Overview of the relative fitness assay. L4 animals grown under auxin (+) or control (ct) conditions carrying a pharyngeal marker in either red (*Pmyo-2::mCherry*) or green (*Pmyo-2::GFP*) are transferred in equal amounts to a single plate. Worms are grown until the plate is full or starved, after which a random subset of worms are transferred to a new plate, effectively diluting the population, after which the proportion of *mCherry+* and *GFP+* progeny is counted. In replicate experiments, pharyngeal markers were swapped for the auxin and control conditions. Reproductive fitness measurements shown in panels B and C were made after one plate dilution, whereas reproductive fitness measured over consecutive plate dilutions is shown in panel G. **(B, C)** Boxplots showing the normalized proportion of progeny originating from worms with a wild-type binucleated (ct, *n* = 40 plates/280 worms and 34 plates/238 worms) or mononucleated (+, *n* = 40 plates/280 worms and 34 plates/238 worms) intestine in relative fitness assays using either the *AID::cdk-1* (B) or *AID::knl-1* alleles (C) to block binucleation, in 2 replicate experiments. Measurements were made after one plate dilution. Boxplots indicate the median and 25th to 75th percentile, error bars indicate min to max values, and individual values are shown as dots. **(D)** Boxplot showing total brood size of *Pges-1::TIR1; AID::knl-1* animals grown under control conditions (ct, *n* = 43 plates) or in the presence of auxin (+, *n* = 43 plates), in 3 replicate experiments. Amounts of progeny were counted for 3 days of egg laying. Boxplots indicate the median and 25th to 75th percentile, error bars indicate min to max values, and individual values are shown as dots. **(E)** Violin boxplots depicting progeny growth rates from worms with increasing maternal age, grown under control conditions (ct, *n* = 9 plates) or in the presence of auxin (+, *n* = 9 plates), in 3 replicate experiments. For each condition, the timing of the L3 molt was quantified (see Methods for details), and the average timing in the control condition for each maternal age was used to calculate the developmental delay. Horizontal lines indicate the median and 25th to 75th percentile, violin plots extend to min and max values and individual values are shown as dots. **(F)** Scatter plots of brood size in the

second generation of *Pges-1::TIR1; AID::knl-1* animals (F2) that originated from animals grown under control (ct, *n* = 70 plates) or auxin (+, *n* = 71 plates) conditions. Amounts of animals were counted for 3 days of egg laying. Line and error bars represent mean and SEM. **(G)** Exponential growth model assuming a intergenerational (green) or transgenerational (orange) effect of blocking binucleation on reproductive fitness. Measurements of the mean reproductive fitness that were obtained over consecutive plate dilutions are shown in black, and error bars depict SEM. *P* values were calculated by Mann–Whitney test. NS, not significant (*P* > 0.05). Underlying data can be found in S1 Data. AID, auxin-inducible degron.

worms lacking the *AID::knl-1* or *AID::cdk-1* alleles did not show developmental delays or reproductive defects (S5D and S5E Fig). Moreover, animals derived from mothers with mononucleated intestines showed a mild decrease in brood sizes compared to controls (Fig 5F). To test whether these developmental effects could account for the observed decreases in fitness, we used an exponential growth model to predict the relative fitness of worms with a mononucleated intestine based on our growth and reproduction measurements. In this model, we differentiated the possibilities of an intergenerational effect of blocking binucleation in the intestine, where only the first generation of progeny is affected, and a transgenerational effect that lasts for several generations. Comparing this model with our experimental data revealed 2 things. First, the plateau in the proportion of progeny coming from mothers with a mononucleated intestine indicates that there is an intergenerational rather than a transgenerational effect on progeny fitness (Fig 5G). Second, the fitness decrease that we measured closely matches the model, suggesting that decreases in progeny growth and reproduction fully explain the difference in relative fitness. Taken together, our data show that intestinal binucleation is important for reproductive fitness and that blocking binucleation decreases progeny growth and reproduction.

## Binucleation of the *C. elegans* intestine is important for the rapid up-regulation of gene expression

To understand why mononucleated cells have decreased levels of *vit-2* gene expression compared to binucleated cells, we performed in-depth analysis of our single-worm RNA sequencing data to identify similarities between genes that are affected by binucleation. First, we found no correlation between expression levels and differential gene expression in animals with a mononucleated intestine, indicating that binucleation is not solely important for the expression of highly or lowly expressed genes (S3A and S3C Fig). Next, we used sequencing data from wild-type worms at different stages of development, available from ModEncode [38], and analyzed the developmental expression profiles of the genes that we identified in our sequencing of animals with mononucleated intestinal nuclei. Overall, genes that are down-regulated in worms with a mononucleated intestine show an increase in gene expression throughout wild-type development, with a strong increase in expression from the L4 to adult stage (Fig 6A). In contrast, genes either unaffected by perturbation of binucleation or genes that are intestinally expressed showed a more constant expression profile during wild-type development. In addition, we found a substantial enrichment (*P* = 6.4e-8) of down-regulated genes located on the X chromosome (S6A Fig). The X chromosome–enriched down-regulation is not restricted to vitellogenin genes, of which 5 out of 6 are located on the X chromosome, but rather a global repression of all X chromosomal genes (S6B Fig). Possibly, an altered X chromosome organization within the nucleus contributes to the decreased expression in mononucleated cells.

To test whether binucleation is important for the rapid up-regulation of gene expression in general, or specifically for expression of genes that are up-regulated upon adulthood, we measured the up-regulation of a heat shock–inducible GFP-NLS reporter in animals with mononucleated or binucleated intestines. In these experiments, L4 stage animals were heat shocked, and nuclear GFP intensities were measured at multiple time points after heat shock (Fig 6B

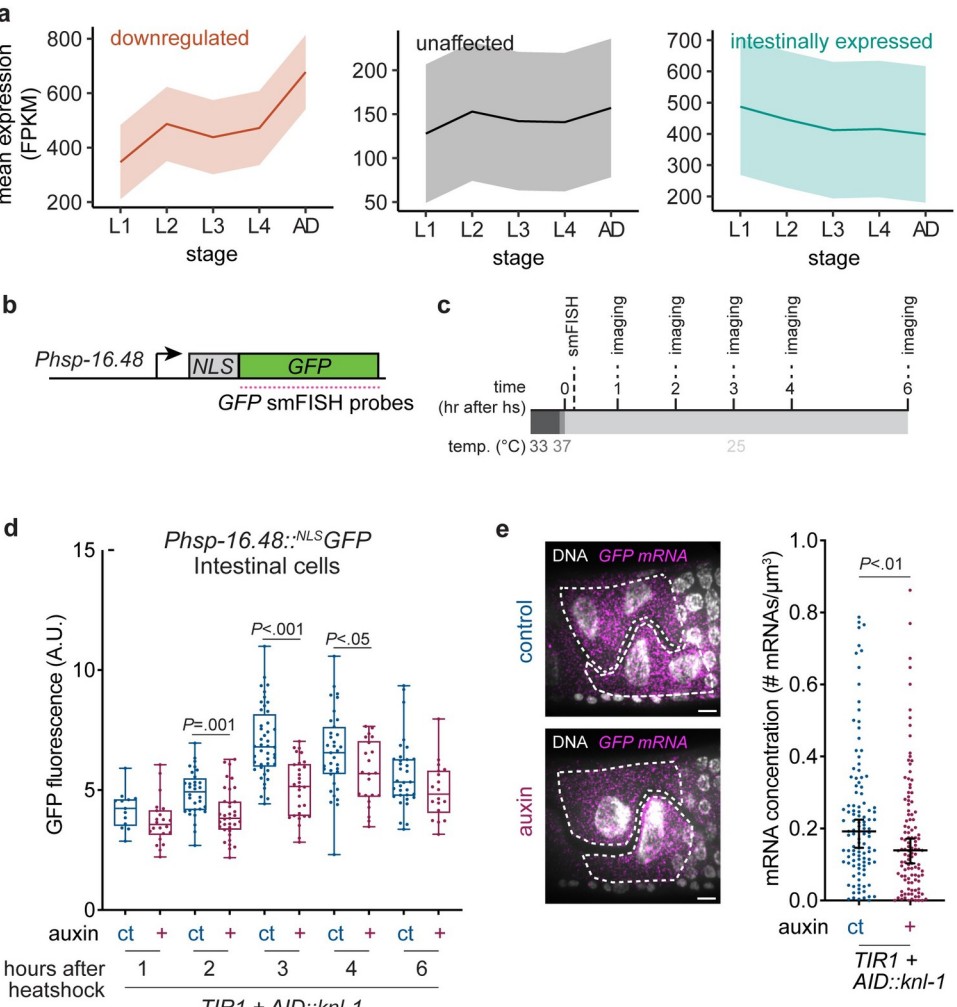

**Fig 6. Binucleation allows rapid up-regulation of transcription. (A)** Mean expression for down-regulated ($n = 96$ genes), unaffected ($n = 113$ genes), or intestinally expressed ($n = 80$ genes) subsets of genes in different developmental stages from ModEncode FPKM gene expression data. Area around curve shows confidence interval. **(B)** Schematic overview of heat shock–inducible NLS-GFP transgene. smFISH probes were designed against the sfGFP sequence. **(C)** Schematic overview of smFISH and imaging experiments following heat shock induction in *Pges-1::TIR1; AID::knl-1* animals. Young L4 stage animals grown under control or auxin conditions were heat shocked in a water bath at 33˚C for 30 minutes, followed by 5 minutes in a 37˚C air incubator. Animals were left to recover at 25˚C. Samples were taken after 15 minutes for smFISH analyses or every hour, for the analysis of nuclear GFP accumulation. **(D)** Total nuclear fluorescence intensities of intestinal nuclei at different time points after heat shock for animals grown under control (ct, $n = 13$ to 44) or auxin (+, $n = 18$ to 35) conditions, in 3 replicate experiments. Boxplots indicate the median and 25th to 75th percentile, error bars indicate min to max values, and individual values are shown as dots. *P* values were calculated by Student *t* test. **(E)** Representative smFISH images and quantifications of cellular *GFP* mRNA concentration in animals grown under auxin (+, $n = 113$) or control (ct, $n = 112$) conditions 15 minutes after heat shock, in 2 replicate experiments. Scatter plot with median (middle line) and 95% confidence interval (error bars). Each dot represents mRNA concentration in a single cell. *P* value was calculated by Mann–Whitney test. Scale bar represents 5 μm. Underlying data can be found in S1 Data. AID, auxin-inducible degron; FPKM, fragments per kilobase of exon per million mapped fragments; smFISH, single molecule fluorescence in situ hybridisation.

and 6C). Interestingly, mononucleated intestinal cells showed a delay in the up-regulation of nuclear GFP after heat shock compared to binucleated controls, whereas the accumulation of GFP signal in body wall muscle nuclei was similar between the 2 conditions (Fig 6D, S7 Fig). To confirm that this delay in up-regulation of heat shock–inducible expression arises at the transcriptional level, we quantified the cellular *GFP* mRNA density 15 minutes after heat

shock induction using smFISH. Consistent with a delay in GFP up-regulation, we observed a significant decrease in *GFP* mRNA levels in mononucleated compared to binucleated intestinal cells (Fig 6E). Together, these results show that rapid up-regulation of transcription is impaired in cells with one rather than two nuclei.

## Increasing vitellogenin levels can rescue the defects in the progeny of animals with mononucleated intestinal cells

Our results suggest that animals with mononucleated intestinal cells have a defect in the rapid up-regulation of vitellogenin gene expression at the L4-to-adult transition, leading to reduced lipid loading in their progeny and reduced fitness. To test whether reduced vitellogenin expression can phenocopy the loss of endomitosis, we generated knockout mutants in *vit-5* and *vit-6*, 2 of the 6 vitellogenin genes. Single and double *vit-5* and *vit-6* mutants had lower brood sizes, and brood sizes were further reduced in the progeny of animals with mononucleated intestines (Fig 7A). This is consistent with previous studies, where it was found that depletion of the yolk uptake receptor RME-2 results in reduced fertility [33,35]. Thus, reduction of yolk proteins gives rise to similar defects as the loss of endomitosis.

We next tested whether transient activation of vitellogenin transcription earlier in development is sufficient to mitigate the differences in *vit-2* expression between binucleated and mononucleated intestines and can rescue the defects that we observe in the progeny of animals with mononucleated intestinal cells. During wild-type development, vitellogenin expression is up-regulated at the L4-to-adult transition by the interaction between transcription factors CEH-60 and UNC-62 [39,40]. We found that intestine-specific overexpression of CEH-60 and UNC-62 was sufficient to activate expression of our vitellogenin reporter as early as the L3 stage, whereas no activation was observed at this stage in control animals (S8 Fig). Furthermore, animals overexpressing CEH-60 and UNC-62 showed an increase in *Pvit-2* expression and their progeny had inherited increased levels of lipids (Fig 7B and 7C), consistent with the function of CEH-60 and UNC-62 to drive vitellogenin transcription and promote lipid loading into oocytes. Importantly, overexpression of CEH-60 and UNC-62 resulted in similar *Pvit-2* expression levels between mononucleated and binucleated intestines, as well as similar levels of lipids in embryos derived from mothers with mononucleated or binucleated intestines (Fig 7B and 7C). These results suggest that CEH-60 and UNC-62 are limiting for vitellogenin expression in mononucleated intestinal cells and that overexpression of these factors is sufficient to rescue the defects in lipid loading in the progeny of animals with mononucleated intestinal cells.

Finally, we tested whether overexpression of CEH-60 and UNC-62 was sufficient to rescue the reduced brood sizes of progeny derived from animals with mononucleated intestines. Although overexpression of CEH-60 and UNC-62 lead to smaller broods compared to wild type, we no longer found differences in the brood sizes of the progeny of animals with either binucleated or mononucleated intestinal cells (Fig 7D). Thus, increasing vitellogenin levels by overexpression of CEH-60 and UNC-62 is sufficient to abrogate the phenotypic consequences of reduced transcription in animals with mononucleated intestines. Taken together, these results indicate that binucleation of the intestine is needed to facilitate rapid transcriptional up-regulation of vitellogenin expression at the L4-to-adult transition, which is important to ensure abundant lipid loading into developing oocytes and promote reproductive fitness.

## Discussion

Our study demonstrates that binucleation is important for the rapid transcriptional up-regulation of gene expression at the onset of adulthood, indicating that the transcriptional output of

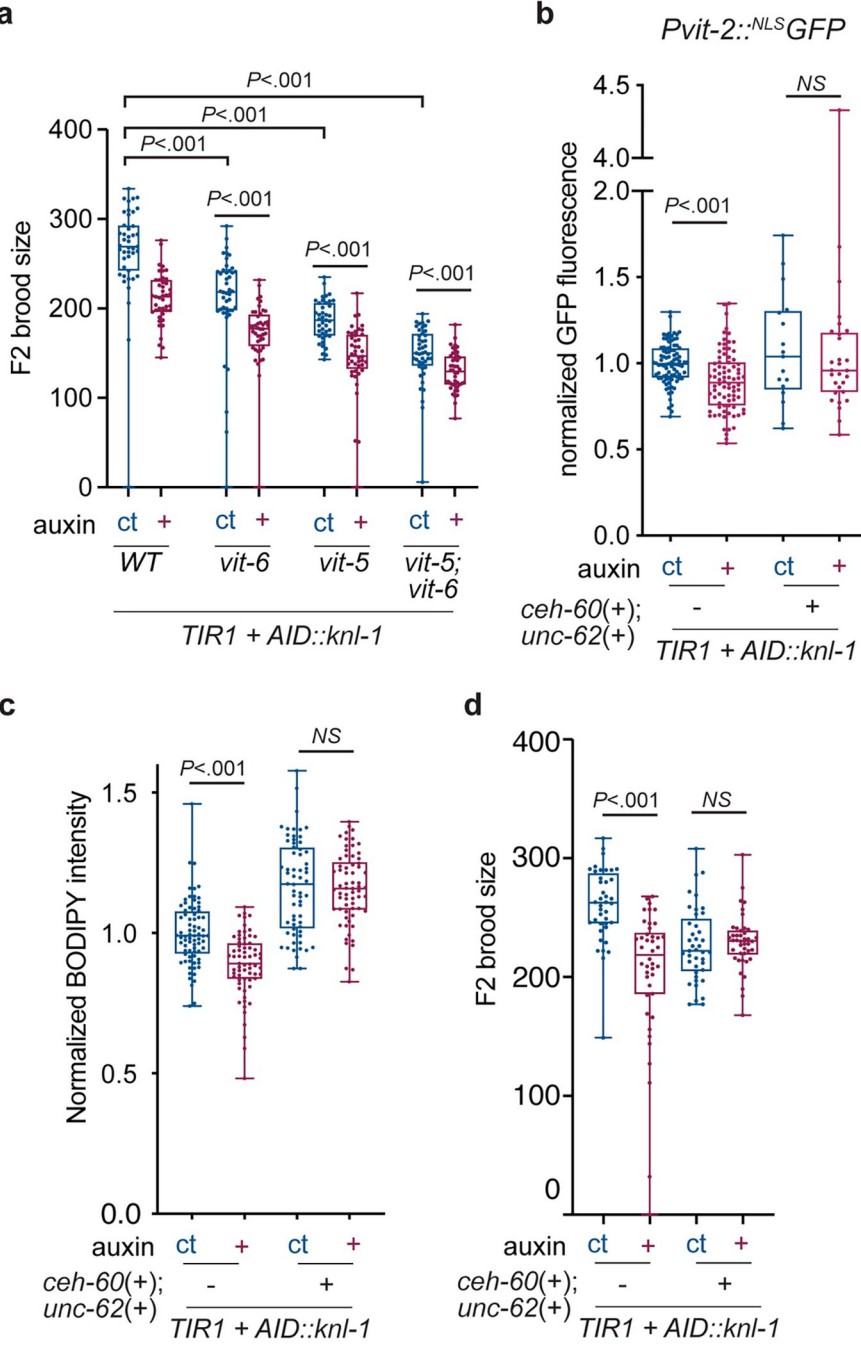

**Fig 7. Transient activation of vitellogenin expression rescues defects of animals with mononucleated intestines.**
**(A)** F2 brood sizes of *Pges-1::TIR1; AID::knl-1* animals containing deletions in either *vit-5*, *vit-6*, or both. F2 brood sizes were measured from the progeny of animals with either mononucleated (+, *n* = 43 to 44) or binucleated (ct, *n* = 41 to 44) intestines that had been laid during the first day of adulthood (maternal age 48–72), in 2 replicate experiments. Animals without *vit-5* or *vit-6* deletions (wild type) were taken along as controls. **(B)** Normalized total *Pvit-2::*$^{NLS}$*GFP* fluorescence intensity per animal, for animals grown under auxin (+) or control (ct) conditions and with (+, *n* = 17 and 29) or without (−, *n* = 82 and 89) intestinal overexpression of transcription factors *ceh-60* and *unc-62*, measured at young adulthood in 5 replicate experiments. Scatter plot with line and error bars indicating mean and SEM. **(C)** Normalized total fluorescence intensities of BODIPY lipid staining of early embryos isolated from young adult (72 hours) *Pges-1::TIR1; AID::knl-1* animals with or without intestinal overexpression of CEH-60 and UNC-62, grown under control (ct, *n* = 76 and 64) or auxin (+, *n* = 67 and 64) conditions in 3 replicate experiments. Boxplots indicate the median and 25th to 75th percentile, error bars indicate min to max values, and individual values are shown as dots. **(D)** F2 brood sizes of *Pges-1::TIR1; AID::knl-1* animals with or without intestinal overexpression of CEH-60 and UNC-

62. F2 brood sizes were measured from the progeny of animals with either mononucleated (+, $n$ = 42 and 42) or binucleated (ct, $n$ = 40 and 41) intestines that had been laid during the first day of adulthood, in 2 replicate experiments. $P$ values were calculated by Mann–Whitney test. NS, not significant ($P > 0.05$). Underlying data can be found in S1 Data. AID, auxin-inducible degron.

a cell is not only determined by the copy number of a gene, but also by whether the DNA is present in a single nucleus or partitioned over 2 or more nuclei. This finding identifies a key role for endomitosis and multinucleation and highlights the importance of genome organization for fine-tuning gene expression control.

Multinucleated cells are present in many different animal tissues; however, it is largely unknown why certain cells become multinucleate and how this influences gene expression. Recent studies have begun to investigate the transcriptional activities of multinucleated cells such as mouse cardiomyocytes, hepatocytes, and skeletal muscle cells [16,41–44]. These analyses have revealed gene expression changes that occur in multinucleated cells; however, it is unclear from these studies whether these transcriptional changes are a direct consequence of multinucleation or arise in response to differentiation cues and are thus only indirectly associated with multinucleation. For example, binucleate cardiomyocytes were shown to have reduced expression of cell cycle genes and an increased expression of genes involved in cardiomyocyte maturation, suggesting that multinucleated cells proliferate less and are more differentiated [16]. Nonetheless, it is unclear whether multinucleation per se influences the expression of cardiomyocyte maturation genes. This could be tested by comparing gene expression profiles between multinucleated 4N cells and mononucleated 4N cells, which is technically challenging because the majority of polyploid murine cardiomyocytes are multinucleated [16].

To understand the transcriptional function of multinucleation for polyploid cells, we developed 2 complementary methods to inhibit endomitosis in *C. elegans* intestines. By specifically depleting either KNL-1 or CDK-1 during intestinal endomitosis, we could directly compare binucleated and mononucleated intestinal cells with the same genome copy number. We found that inhibition of endomitosis by depleting either KNL-1 or CDK-1 gave very similar phenotypes and resulted in the down-regulation of genes that are required for the transition from larval development to adulthood. Surprisingly, we found that restricting all DNA copies into one nucleus instead of two does not influence cell size or intestinal morphology, but specifically affects the efficiency of fast-acting transcriptional responses. It is possible that differences in nuclear volume or nuclear surface-to-volume cause the transcriptional defects that we observed in mononucleated cells. Changes in the surface-to-volume ratio could influence nuclear import and export dynamics of transcriptional regulators, leading to altered transcriptional dynamics. An increased nuclear volume could lead to lower concentrations of transcription factors in the nucleoplasm, reducing gene target engagement and transcription output. Finally, it has been shown that the expression of developmentally regulated genes often correlates with a specific subnuclear localization and that interactions with the nuclear lamina, the nuclear envelope or the nucleolus can influence transcriptional activity [45–47]. Therefore, changes in the nuclear surface-to-volume ratio and increases in nuclear volume could influence transcription by a differential distribution of loci within the nucleus.

In this study, we show that in *C. elegans* intestinal cells binucleation is important for the rapid up-regulation of vitellogenin genes in young adult animals. We find that reduced expression of vitellogenins in mononucleated intestinal cells leads to progeny with developmental delays and reduced fitness. Two transcription factors, CEH-60 and UNC-62, have been previously shown to drive the large-scale gene expression changes that occur at the transition from

larval development to adulthood [39]. Depletion of UNC-62 or CEH-60 decreases the expression of vitellogenins in the intestine, leading to reduced fat and lipid levels in embryos and reduced fitness of the progeny [36,39,40,48]. Thus, it is possible that reduced transcriptional activity of UNC-62 or CEH-60 underlies the fitness defects that we observe in the progeny of animals with mononucleated intestines. Consistent with this hypothesis, we find that overexpression of CEH-60 and UNC-62 is sufficient to rescue the reproductive fitness and oocyte lipid loading defects that we observe upon inhibition of endomitosis. These results suggest that impaired CEH-60/UNC-62 activity underlies the defects of animals with mononucleated intestinal cells. Interestingly, in addition to regulating vitellogenin gene expression, UNC-62 also binds the promoters of *asp-6*, *ets-4*, *cpl-1*, *cpr-4*, *spp-5*, and *ZK1320.3* [39], 6 other genes that we found to be down-regulated in animals with mononucleated intestinal cells. Depletion of *ets-4*, *cpl-1*, and *spp-5* has been shown to give rise to defects in reproductive fitness [49–51]; thus, it is possible that increasing their levels by overexpression of CEH-60 and UNC-62 is contributing to the rescue in brood size and lipid loading in the progeny of animals with mononucleated intestinal cells.

In future work, it will be interesting to understand how CEH-60 and UNC-62 could be acting differently in mononucleated or binucleated cells. From our RNA sequencing experiments, we find that *unc-62* and *ceh-60* are not differentially expressed between mononucleated and binucleated cells, suggesting that differences in CEH-60/UNC-62 activities arise posttranscriptionally. Interestingly, in mononucleated cells, we also find reduced expression of the ribosomal subunit *rpl-11.2*, and the transcriptional activator *ets-4*, which functions in a variety of different tissues and contexts and also controls the expression of *vit-2*, *vit-3*, *vit-4*, *vit-5*, and *ceh-60* [51]. Thus, it is possible that lower levels of *rpl-11.2* or *ets-4* are leading to a global reduction in protein translation or RNA polymerase II-specific transcription in mononucleated cells. Another possibility is that the nuclear accumulation of transcription factors such as UNC-62 and CEH-60 is less efficient in mononucleated cells, resulting in reduced transcription. Interestingly, when we compared the transcriptional activation of a heat shock reporter driving GFP expression in animals with either binucleated or mononucleated intestinal cells, we found that mononucleated cells display a delay in the accumulation of both *gfp* mRNA and protein. These results suggest that binucleation is important for the rapid up-regulation of gene expression in general, likely by promoting transcriptional activity, and perhaps also by increasing protein translation, which remains to be tested. Of note, the transcriptional activation experiments were only performed in the *AID*::*knl-1* strain due to technical limitations of the weakly hypomorphic *AID*::*cdk-1* strain; thus, we cannot rule out that the transcriptional defects that we observe are specific to the depletion of KNL-1.

Taken together, this work sheds light on the function of multinucleation in polyploid cells, revealing that the packaging of nuclear DNA into multiple nuclei can be beneficial to facilitate rapid transcriptional responses. As multinucleated cells are common in many animal and plant tissues, and can also arise in diseases such as cancer, Alzheimer, and during viral infections [52–54], our findings raise the possibility that multinucleated tissues function to maximize transcriptional activities in many contexts.

## Methods

### Worm culture

Worms were cultured on nematode growth medium (NGM) plates seeded with OP50 *Escherichia coli* bacteria according to standard protocols. All strains were maintained at 20°C, unless indicated otherwise. N2 Bristol were used as wild types. For auxin plates, NGM was supplemented with 0.5 mM auxin (I2886; Sigma-Aldrich, St. Louis, MO, USA) after autoclaving and

cooling to <55˚C. As auxin is dissolved in 100% ethanol, control plates are supplemented with equal volumes of 100% ethanol without auxin. For auxin experiments, worms were synchronized by isolation of eggs from adult hermaphrodites through alkaline hypochlorite treatment and hatching overnight in M9 medium in the absence of food. Next, synchronized L1 larvae were divided over 2 conditions: auxin (+) and auxin control (ct). For the auxin condition, worms were grown on auxin for 24 hours, after which they are transferred to control plates. Auxin-control worms were grown on control plates for 24 hours and transferred to auxin plates from 24 to 48 hours of development. For fitness assays, worms were grown on NGM plates supplemented with 100 mg/mL ampicillin (Sigma-Aldrich A-9518) L4440 *E. coli* bacteria to prevent contamination. For heat shock experiments, animals were heat shocked in a water bath at 33˚C for 30 minutes, followed by 5 minutes in a 37˚C air incubator. Animals were then transferred to 25˚C, and samples were taken after 15 minutes for smFISH analysis (see below) or every hour, for the analysis of nuclear GFP accumulation.

## Strains

*C. elegans* strains used in this study are described in S1 Table. CDK-1 and KNL-1 N-terminal AID tags were inserted into the CA1209 strain using CRISPR/Cas-9 genome editing with 2 overlapping ssODNs (IDT 4 nmole Ultramers) as template for homology-directed repair, as described by Paix and colleagues [55]. *vit-5(mat169)* and *vit-6(mat171)* deletions were generated using CRISPR/Cas-9 genome editing by using two 3′ guides and two 5′ guides and a *rol-6* co-injection marker. CRISPR/Cas-9 insertions and deletions were selected by genomic PCRs and confirmed by DNA sequencing. Single guide RNA and ssODN repair template sequences are listed in S2 Table. Intestinal markers (*matIs53*), a nuclear membrane marker (*matIs104*), pharyngeal markers (*matIs114* and *matIs116*), and an S/G2-phase marker were integrated using gamma irradiation and outcrossed 4 times. The transcriptional *Pvit-2::sfGFP* reporter strain (*matIs137*) was integrated using CRISPR/Cas-9 mediated genome editing at the MosI insertion allele *ttTi5605* on chromosome II. For the intestinal CEH-60 and UNC-62 overexpression experiments, 2 strains were generated: an extrachromosomal overexpression strain, containing *Pelt-2::ceh-60; Pges-1::BFP-P2A-unc-62; Pmyo-2::mCherry* (matEx151, used in Fig 7B), and an integrated overexpression strain, containing *Pelt-2::ceh-60; Pges-1::BFP-P2A-unc-62; Plin-48::TdTomato (matIs172*, used for experiments in Fig 7C).

## Microscopy and image analysis

Static and time-lapse epifluorescence imaging of intestinal fluorescent markers, nuclear markers, chromosome markers, vitellogenin levels, and DNA/lipid staining was performed on a Ultraview Vox spinning disk confocal microscope (Perkin Elmer, Waltham, MA, USA) equipped with a CMOS digital camera (C13440; Hamamatsu Photonics, Hamamatsu-city, Japan). Differential interference contrast (DIC) imaging of worm size was performed using a 20× objective on a Axio Imager M2 microscope (Carl Zeiss, Oberkochen, Germany) equipped with an Axiocam camera (Carl Zeiss, Oberkochen, Germany). Imaging of cell cycle progression using the S/G2-phase marker was performed on a DM6000 Fluorescence Microscope (Leica Microsystems GmbH, Wetzlar, Germany). For imaging, animals were mounted onto 2% agarose pads (adults) or 7% agarose pads (L1 and L2 larvae) and immobilized with 10 mM sodium azide for static imaging, or 1 ng/mL levamisole and 100 ng/mL tricaine for time-lapse imaging. Young adult intestinal markers or nuclear membrane markers were imaged with a 20× or 63× objective, using 2× binning. Chromosome markers in L2/L3 animals were imaged with a 100× objective, without binning. Time-lapse imaging of intestinal markers during endomitosis in L1 larvae was performed with a 100× objective, with 2× binning. For imaging in the

presence of auxin, M9 and agarose pads were supplemented with 0.5 mM auxin. To measure vitellogenin and lipid levels, embryos were imaged at 100× magnification, using 2× binning. PI DNA staining was imaged with a 63× objective, with 2× binning.

Images were processed and analyzed using Fiji software [56]. For quantifications of intestinal cell and nuclei numbers, images of young adults (64 hours) were taken using a 20× objective lens and stitched together using the pairwise stitching tool from Fiji [57]. For quantifications of PI stainings, GFP-VIT-2 fluorescence intensity, and BODIPY stainings, z-axis serial scans distanced at 50% the optical slice depth were summed in a z-stack projection after background subtraction, after which the integrated density of the region of interest was measured. For quantifications of nuclear GFP after heat shock induction, 20 z-slices with 0.5 μm spacing (corresponding to 50% of the optical slice depth) were summed in a z-stack projection. In this case, no background subtraction was performed, as background intensity analysis showed similar background levels between conditions. From the z-projections, integrated densities of the nuclear regions were measured. For binucleated cells, the integrated densities of the 2 nuclei were summed.

## DNA staining

For PI quantitative DNA staining, freeze-cracked young adult hermaphrodites were fixed with Carnoy's solution and treated with RNAse A (Invitrogen #12091021; Thermo Fisher Scientific, Waltham, MA, USA) as previously described [58]. PI staining was performed using 100 μg/mL PI (#P4170; Sigma-Aldrich, St. Louis, MO, USA) for 90 minutes at 37°C, after which slides were washed 3 times and mounted using ProLong gold antifade mountant (Invitrogen #P36934).

## Competitive fitness assay and modeling

Competitive fitness assays were performed using the *AID*::*cdk-1* strains GAL182 (with *Pmyo-2*::*GFP*) and GAL191 (with *Pmyo-2*::*mCherry*) and the *AID*::*knl-1* strains GAL160 (with *Pmyo-2*::*mCherry*) and GAL162 (with *Pmyo-2*::*GFP*). Prior to the start of the assay, worms were grown under auxin or auxin-control conditions as described above. Each experiment contained 3 conditions: control-GFP versus control-mCherry, auxin-GFP versus control-mCherry, and control-GFP versus auxin-mCherry. In the first condition, control-GFP versus control-mCherry, auxin-control worms with a green fluorescent pharyngeal marker were grown on the same plate as auxin-control worms with a red fluorescent pharyngeal marker. Second, for the auxin-GFP versus control-mCherry condition, auxin worms with a green fluorescent pharyngeal marker were grown on the same plate as auxin-control worms with a red fluorescent pharyngeal marker. Finally, in the control-GFP versus auxin-mCherry condition, auxin-control worms with a green fluorescent pharyngeal marker were grown on the same plate as auxin worms with a red fluorescent pharyngeal marker. For each of the conditions, 10 replicates were performed. In each replicate, 7 worms of both conditions were transferred to single plates, which were left to grow until starvation (for *AID*::*knl-1* worms) or until the plate was close to starvation (for *AID*::*cdk-1* strains, as these were prone to become sterile upon starvation). Each plate was chunked to 2 new plates, one of which was grown for 1 day and used to count relative amounts of GFP+ and mCherry+ progeny, whereas the other plate was used to initiate another growth cycle. For counting, animals were washed off plates, mounted on 2% agarose pads containing 10 mM sodium azide, and counted using a Zeiss Axioscope. Relative amounts of control-GFP versus control-mCherry progeny were used to normalize relative amounts of auxin-GFP versus control-mCherry and control-GFP versus auxin-mCherry progeny.

For modeling, the average number of eggs and average time of egg laying were calculated for the first and second generation, using a predicted Gaussian distribution curve on measurements of eggs laid per day. These numbers were used to calculate an exponential growth rate $\lambda$, equal to $\frac{\ln(x)}{t}$, where $x$ is the average number of eggs that one animal produces, and $t$ is the average time (in days) at which these eggs are laid. The predicted relative fitness change per day of growth was determined by the ratio of the growth rate of worms with a mononucleated intestine divided by the growth rate of worms with a binucleated intestine. In the model assuming a transgenerational effect, this change in relative fitness is constant for several generations. When assuming a intergenerational effect, the change in relative fitness is set to zero from the start of generation F2.

## Single-worm RNA sequencing and analysis

Transgenic animals were grown until adulthood under auxin or control conditions. Young adult hermaphrodites were picked using tweezers and washed in 0.5-mL sterile demineralized water before adding 200-μL trizol (Invitrogen 15596018). Samples were stored at −80˚C for a limited time. mRNA extraction, barcoding, reverse transcription, in vitro transcription, and Illumina sequencing library preparation were performed according to the robotized version of the CEL-seq2 protocol [59] using the SuperScript II Double-Stranded cDNA synthesis kit (Thermo Fisher Scientific, Waltham, MA, USA), Agencourt AMPure XP beads (Beckman Coulter, Brea, CA, USA), and randomhexRt for converting aRNA to cDNA using random priming. The libraries were sequenced paired-end at 50 bp read length on an Illumina HiSeq 2500. The 50 base pair paired-end reads were aligned to the *C. elegans* reference transcriptome compiled from the *C. elegans* reference genome WS249. Raw data were processed using R (v3.6.2). Samples were filtered to include those with sufficient counts per transcript and gene, results were filtered for batch consistency, and only genes that were found at least once in every sample were included for downstream analysis (S3 Fig). Further filtering of genes was performed to include only genes for which intestinal expression has been annotated in previous analyses of intestinal gene expression [60–63]. Additionally, possibly growth-delayed or sterile outlier animals were removed from the dataset by analyzing the expression of spermatogenesis and oogenesis genes as described by Perez and colleagues [33]. For this, genes associated with Gene Ontology (GO) terms for oogenesis ($n = 50$ genes) and spermatogenesis ($n = 57$ genes) were used to identify outlier animals, and normalized read counts were summed per animal and plotted to identify outlier animals. Presence of outlier animals with very low levels of oogenesis and spermatogenesis gene expression was found to be independent of batch, condition, or genetic background. This resulted in a dataset consisting of 143 animals with gene expression data for a total of 2,638 genes for *AID*::*knl-1* animals and 2,724 genes for *AID*::*cdk-1* animals. The DEseq R package was used to perform differential gene expression analysis. Genes with an adjusted *P* value below 0.05 and a log2 fold change of gene expression within the top 25% (corresponding to an absolute log2 fold change above 0.514 in *AID*::*knl-1* animals and 0.162 in *AID*::*cdk-1* animals) were considered to be significantly and substantially differentially expressed. GO term enrichment analysis was performed using the InterMineR package for R. Fragments per kilobase of exon per million mapped fragments (FPKM) gene expression data from different developmental stages were retrieved from ModEncode through WormBase [64] and analyzed in R. A subset of intestinally expressed genes was composed by combining lists of genes that were identified as intestinally unique, enriched, or expressed in previous expression studies [60–63]. A Venn diagram was constructed using the BioVenn web application [65]. The RNA-sequencing data have been deposited at the Gene Expression Omnibus (GSE169330).

## Lipid staining

Embryo lipid staining was performed as described previously [33]. In short, embryos were obtained by alkaline hypochlorite treatment of adult hermaphrodites, transferred to PCR tubes (Eppendorf, Hamburg, Germany) containing 4% formaldehyde in M9 buffer and subjected to 3 freeze-thaw cycles in liquid nitrogen and a 37˚C water bath. Embryos were then incubated with 1 μg/ml BODIPY 493/503 (Thermo Fisher Scientific) in M9 buffer for 1 hour and washed 3 times with M9 + 0.01% Triton X-100 (Sigma #9001-93-1). Next, nuclei were stained with 5 ng/mL DAPI and imaged by epifluorescence microscopy.

Adults were fixed for lipid staining by incubating in 4% paraformaldehyde (Sigma-Aldrich 158127) for 1 hour at room temperature. Adults were then washed twice with PBS, resuspended in 95% ethanol and washed again in PBS. Next, fixed animals were incubated with 1 μg/ml BODIPY 493/503 (Thermo Fisher Scientific) in M9 buffer for 1 hour and washed 3 times with M9 + 0.01% Triton X-100. Animals were stained with 5 ng/mL DAPI and imaged by epifluorescence microscopy.

## Viability assay

Worms were grown until the L4 stage under auxin or auxin-control conditions as described above. Single hermaphrodites were then transferred to new plates every 24 hours. On each day, the number of live progeny and unhatched eggs were counted 24 to 48 hours after removal of the adult worm to calculate the total brood size and embryonic lethality.

## Progeny growth assay

Animals were grown until young adults under auxin, control, or no auxin condition as described above. Developmental transition timings were performed as previously described using a wash-off staging [33]. Briefly, plates containing synchronized gravid hermaphrodites were washed to remove all animals except embryos that remained attached to the solid media. To tightly synchronize a population of animals, larvae that had hatched within 1 hour after wash-off were collected and transferred to new plates. The fraction of animals that had undergone the transition from L3 to L4 was counted for 3 replicate plates containing around 150 worms each, in 3 separate experiments. To estimate the time at which half of the population had undergone the developmental transition, each population was scored at least twice, once before and once after half of the population had undergone the L4 transition. Count data were modeled using a binomial generalized linear model, and the time at which half of the population had undergone the developmental transition ($T_{50}$) was estimated using the function "dose.p" of the library "MASS" in R. Moreover, 95% confidence intervals were estimated by multiplying the standard error obtained by the "dose.p" function by 1.96. The significance of the difference between 2 calculated $T_{50}$ values was determined using the "comped" function in the library "drc" in R, which takes as input the 2 $T_{50}$ values, their standard error and a desired probability level (such as 0.95).

## Single molecule fluorescence in situ hybridization (smFISH)

smFISH was performed as previously described [66] with some minor adjustments. In short, animals were fixed using 4% paraformaldehyde, resuspended in 70% ethanol, and stored at 4˚C for up to 2 weeks. Short oligonucleotide probes complementary to the *sfGFP* and *vit-2* sequences (see S3 Table for sequences) were designed using a web-based algorithm (www.biosearchtech.com/stellaris-designer) and ordered with a 3-amino modification to enable coupling to Cy5 (#PA25001; GE Healthcare, Chicago, IL, USA) as previously described [67].

Hybridization was performed overnight at 37°C in the dark, after which samples were washed and stained with DAPI. Z-stacks of Int3A and Int3V cells were generated using spinning disk microscopy and mRNA molecules were counted using the batch analysis mode in FISH-quant [68]. To quantify *sfGFP* mRNA densities, 4.5-μm substacks were made in Fiji of each cell centered around the nucleus/nuclei. From these substacks, mRNA spots were counted in 3 regions of 4.55 μm × 4.55 μm, and spot density was calculated as # of spots per μm$^3$. To quantify the number of *vit-2* mRNAs, substacks were made in Fiji of each cell, using the GFP-PH marker to determine cell boundaries. Cell outlines, as well as nuclei and nascent transcripts, were then annotated in FISH-quant and counted using the batch analysis mode. Nascent *vit-2* transcripts were defined as nuclear smFISH spots that were at least 200 nm in XY and 300 nm in Z dimension and that contained at least 2 mRNAs, which was calculated by averaging the cytoplasmic mRNA spots and comparing the amplitude of the annotated transcription sites to individual mRNA molecules, using the transcription site quantification interface in FISH-quant.

## Statistics and reproducibility

The sample size *(n)* as well as the number of replicate experiments performed for each experiment is described in the corresponding figure legend. All statistical analyses were performed using Prism version 8.4.3 (Graphpad, San Diego, CA, USA) except for sequencing data analyses. Quantitative data displayed as boxplots indicate the median and 25th to 75th percentile, error bars indicate min to max values, and individual values are shown as dots, unless indicated otherwise. Two-tailed unpaired Student *t* tests were used for pairwise comparisons between groups with similar Gaussian distributions. Mann–Whitney U tests were used for pairwise comparisons between groups with non-Gaussian distributions. The type of statistical test performed is indicated in the corresponding figure legends. Statistical significance values *(P)* are shown directly within the figure above pairwise comparisons.

## Supporting information

**S1 Fig. Perturbation of binucleation through degradation of KNL-1 prevents DNA segregation and does not affect subsequent S-phase timing. (A)** Stills of time-lapse videos of cells undergoing endomitosis in the absence (control) or presence of auxin, showing intestinal H2B-mCherry (DNA, shown in magenta) and GFP-PH (membrane, shown in green). Scale bar is 2 μm. Time stamp indicates time after nuclear envelope breakdown. **(B)** Quantification of mitotic timing, the duration from nuclear envelope breakdown to nuclear envelope reformation, in the absence (ct, *n* = 4) or presence (+, *n* = 7) of auxin. Bar graph showing mean and error bars indicate SEM. *P* values were calculated by unpaired Student *t* test. **(C)** Overview of LacI/LacO system for detection of individual chromosomes in polyploid cells. A series of LacO repeats present on chromosome V are visualized upon heat shock–induced expression of a LacI fused to GFP. After endomitosis in L1, a binucleated cell with two 2C nuclei or a mononucleated cell with a single 4C nucleus should show 4 individual chromosomes if no segregation errors occurred. If errors did occur during segregation, an alternate number of individual chromosomes should be visible. **(D)** Fluorescent images and violin plots showing chromosome cluster counts of fluorescent LacI::GFP foci in control (ct, *n* = 98) and auxin-treated (+, *n* = 82) worms. To distinguish LacI::GFP signal from cytoplasmic autofluorescent vesicles, only nuclear dots were counted as chromosome clusters. Error bars represent min and max values, and horizontal bars represent median. Scale bar is 5 μm. **(E)** Average percentage of animals in which intestinal cells are undergoing G2 or S phase, determined by the presence of CYB-1$^{DB}$::mCherry, during endomitotic and subsequent endoreplicative cycles in the first larval stage for 60 to 200 worms per condition per time point, in 3 replicate experiments. *x* Axis depicts hours

after starved L1 animals were placed on food at 15 degrees and starts at 16 hours. *P* values were calculated by Fisher exact. Error bars represent standard deviation between replicate experiments, if applicable. Underlying data can be found in S1 Data.
(TIF)

**S2 Fig. Mononucleation results in an altered circumference-to-area ratio. (A)** Schematic depicting the nuclear parameters measured, area and circumference of a nuclear midplane sections. **(B–D)** Boxplots depicting nuclear section area (B), circumference (C) and nuclear circumference-to-area ratio (D) in binucleated (ct, *n* = 60) or mononucleated (+, *n* = 56) cells. Measurements were made at the midplane of the nucleus. Boxplots indicate the median and 25th to 75th percentile, error bars indicate min to max values, and individual values are shown as dots. *P* values were calculated by Mann–Whitney (B, C) and unpaired Student *t* test (D). Underlying data can be found in S1 Data.
(TIF)

**S3 Fig. Overlap in differential gene expression of single young adult worms containing either *AID::knl-1* or *AID::cdk-1*. (A–C)** Volcano and MA plots of RNA sequencing data depicting the transcriptional gene up- and down-regulation in worms with a mononucleated intestine, compared to worms with a binucleated (wild type) intestine in animals containing either *AID::knl-1* (A) or *AID::cdk-1* (B, C), in relation to gene expression levels. Red dots represent genes that are differentially expressed with an adjusted *P* value below 0.05 and belong to the top 25% regarding absolute log2(foldchange). (D, E) Histogram depicting the log2 of read counts per unique molecular identifier (UMI) for each gene found in animals containing either *AID::knl-1* (D) or *AID::cdk-1* (E). (F) Dot plot depicting the correlation between differential expression in *AID::knl-1* (x-axis) and *AID::cdk-1* (y-axis) animals with a mononucleated versus binucleated intestine. Red dots represent genes that are differentially expressed in the combined dataset with an adjusted *P* value below 0.05. Genes significantly differentially expressed in both *AID::knl-1* and *AID::cdk-1* comparisons individually were annotated with their gene name (excluding genes without a gene name). Underlying data are available at the Gene Expression Omnibus, identifier GSE169330, and in S1–S3 Data. AID, auxin-inducible degron.
(TIF)

**S4 Fig. Differential expression of vitellogenin genes in RNA sequencing of animals with a mononucleated versus binucleated intestine. (A–G)** Tukey boxplots showing the separate (A–F) and accumulated (G) RNA expression levels for all 6 vitellogenin genes (*vit-1* through *vit-6*) in auxin-control (ct, *n* = 33 for *AID::knl-1* and n = 39 for *AID::cdk-1*) or auxin-treated (+, *n* = 35 for *AID::knl-1* and *n* = 36 for *AID::cdk-1*). Each dot represents the expression in one worm. Underlying data are available at the Gene Expression Omnibus, identifier GSE169330, and in S1 Data. AID, auxin-inducible degron.
(TIF)

**S5 Fig. Growing worms on auxin in the absence of an AID tag does not affect reproduction or progeny growth. (A)** Overview of experimental procedures for quantifications of brood size. Mixed stage embryos are isolated from adult hermaphrodites containing intestinal expressed TIR1 and either *AID::cdk-1* or *AID::knl-1*, and starved overnight to yield a synchronized population of arrested L1 animals. The population of starved L1 animals is split into 2 conditions: an auxin (+) condition and a control (ct) condition. For the auxin condition, animals are grown on plates containing auxin for the first 24 hours of postembryonic development, when endomitosis normally occurs, and transferred to plates without auxin after this period. Animals in control conditions are grown on plates without auxin for the first 24 hours

of development, and transferred to plates containing auxin for 24 to 48 hours of postembryonic development, when intestinal endomitosis has already occurred and neither KNL-1 or CDK-1 are required in the intestine. Worms are transferred to new plates at 24, 48, 72, 96, and 120 hours. Egg laying starts between 48 and 72 hours of postembryonic development, while virtually no eggs are laid after 120 hours of postembryonic development. After removal of animals from a plate, the remaining eggs are incubated for 16 to 18 hours to allow hatching, before progeny are counted for quantifications of brood size. **(B)** Brood sizes of *Pges-1*::*TIR1* or *Pges-1*::*TIR1; AID*::*cdk-1* animals grown without auxin. **(C)** Brood sizes of *Pges-1*::*TIR1; AID*::*cdk-1* animals grown under control (ct) or auxin (+) conditions. **(D)** Brood sizes of animals expressing *Pges-1*::*TIR1* and grown under control (ct) or auxin (+) conditions. Boxplots indicate the median and 25th to 75th percentile, error bars indicate min to max values, and individual values are shown as dots. *P* values were calculated by Mann–Whitney test. **(E)** Violin box plots depicting progeny growth rates of animals expressing *Pges-1*::*TIR1* that were derived from mothers of different ages, and grown under control conditions (ct, $n = 18$ plates) or in the presence of auxin (ct, $n = 18$ plates), in 3 replicate experiments. Horizontal lines indicate the median and 25th to 75th percentile, violin plots extend to min and max values, and individual values are shown as dots. *P* value was calculated by Mann–Whitney test. Underlying data can be found in S1 Data. AID, auxin-inducible degron.
(TIF)

**S6 Fig. Mononucleation of the intestine causes an X chromosome specific decrease in expression levels. (A)** Tukey boxplots showing the log2 fold change differential gene expression per chromosome in *Pges-1*::*TIR1; AID*::*cdk-1* and *Pges-1*::*TIR1; AID*::*knl-1* animals with a mononucleated versus binucleated intestine. *P* values of the comparison between X chromosomal and autosomal differential gene expression were calculated by Wilcoxon rank sum test. **(B)** Compiled log2 fold change of differential expression of genes located on chromosome V and X in *Pges-1*::*TIR1; AID*::*cdk-1* and *Pges-1*::*TIR1; AID*::*knl-1* animals with a mononucleated versus binucleated intestine. Data are compiled from single-worm RNA sequencing data of both *Pges-1*::*TIR1; AID*::*cdk-1* and *Pges-1*::*TIR1; AID*::*knl-1* strains. Genes with a top 25% percent absolute log2 fold change are indicated in dark gray, vitellogenins are indicated as red dots. Underlying data are available at the Gene Expression Omnibus, identifier GSE169330, and in S1 Data. AID, auxin-inducible degron.
(TIF)

**S7 Fig. Heat shock response in body wall muscle cells is not affected by mononucleation of the intestine.** Boxplots with total nuclear fluorescence intensities of body wall muscle nuclei at different time points after heat shock in auxin-control (ct, $n = 9$ to 23) or auxin-treated (+, $n = 10$ to 20) *Pges-1*::*TIR1; AID*::*knl-1; Phsp-16.48*::$^{NLS}$*GFP* animals, performed in 3 replicate experiments. Boxplots indicate the median and 25th to 75th percentile, error bars indicate min to max values, and individual values are shown as dots. *P* values were calculated by Mann–Whitney test. Underlying data can be found in S1 Data. AID, auxin-inducible degron.
(TIF)

**S8 Fig. Early expression of *ceh-60* and *unc-62* induces vitellogenin promoter activity before adulthood.** Bar graph depicting the percentage of animals with high levels of normalized total GFP fluorescence intensity per animal containing *Pvit-2*::*GFP-NLS*, for L3 animals with (+, $n = 36$) or without (−, $n = 165$) an intestinal overexpression of transcription factors *ceh-60* and *unc-62*. A high level of total GFP fluorescence intensity is defined as more than twice the average control levels of GFP fluorescence intensity after background subtraction. Underlying data

can be found in S1 Data.
(TIF)

**S1 Table. *C. elegans* strains used in this study.**
(PDF)

**S2 Table. Single guide RNA and ssODN repair template sequences.**
(PDF)

**S3 Table. smFISH probe sequences.** smFISH, single molecule fluorescence in situ hybridization.
(PDF)

**S1 Movie. Endomitosis in *Pges-1*::*TIR1*; *AID*::*knl-1* animals grown under control conditions.** AID, auxin-inducible degron.
(AVI)

**S2 Movie. Endomitosis in *Pges-1*::*TIR1*; *AID*::*knl-1* animals grown under auxin conditions.** AID, auxin-inducible degron.
(AVI)

**S1 Data. Data underlying Figs 1E, 1G, 1H, 2A, 2B, 2D, 2E, 2G, 2H, 2K, 2L, 2M, 3A, 3B, 4A, 4C, 4D, 4E, 4F, 4G, 5B, 5C, 5E, 5F, 5G, 6A, 6D, 6E, 7A, 7B, 7C and 7D and S1B, S1D, S1E, S2B, S2C, S2D, S3A, S3C, S3F, S4A, S4B, S4C, S4D, S4E, S4F, S4G, S5B, S5C, S5D, S5E, S6A, S6B, S7 and S8 Figs.**
(XLSX)

**S2 Data. Data underlying S3D Fig.**
(CSV)

**S3 Data. Data underlying S3E Fig.**
(CSV)

# Acknowledgments

We thank members of the Galli, Korswagen, and Kops labs for helpful discussions; Tim Hoek for help and feedback on fitness assay modeling; and Marvin Tanenbaum for critical reading of the manuscript. We also thank Anko de Graaff and the Hubrecht Imaging Center (HIC) for assistance with microscopy. The spinning disk microscope is funded by equipment grant 834.11.002 from the Dutch Organization of Scientific Research (NWO). We thank the Temmerman lab for sharing the plasmid containing *Pelt-2*::*ceh-60*. Some strains were provided by the CGC, which is funded by the NIH Office of Research Infrastructure Programs (P40 OD010440).

# Author Contributions

**Conceptualization:** Lotte M. van Rijnberk, Matilde Galli.

**Formal analysis:** Lotte M. van Rijnberk, Ramon Barrull-Mascaró.

**Funding acquisition:** Matilde Galli.

**Investigation:** Lotte M. van Rijnberk, Ramon Barrull-Mascaró, Reinier L. van der Palen, Matilde Galli.

**Methodology:** Lotte M. van Rijnberk, Ramon Barrull-Mascaró, Reinier L. van der Palen, Erik S. Schild, Matilde Galli.

**Resources:** Hendrik C. Korswagen, Matilde Galli.

**Writing – original draft:** Lotte M. van Rijnberk, Matilde Galli.

**Writing – review & editing:** Lotte M. van Rijnberk, Hendrik C. Korswagen, Matilde Galli.

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
