## [Editor Report · Decision Letter 0]

20 Dec 2021

Dear Dr Galli, 

Thank you for submitting your manuscript entitled "Endomitosis controls tissue-specific gene expression during development" for consideration as a Research Article by PLOS Biology.

Your manuscript has now been evaluated by the PLOS Biology editorial staff, as well as by an academic editor with relevant expertise, and I am writing to let you know that we would like to send your revised manuscript back to the previous Review Commons reviewers. 

However, before we can send your manuscript back out for re-review, we need you to complete your submission by providing the metadata that is required for full assessment. To this end, please login to Editorial Manager where you will find the paper in the 'Submissions Needing Revisions' folder on your homepage. Please click 'Revise Submission' from the Action Links and complete all additional questions in the submission questionnaire.

Once your full submission is complete, your paper will undergo a series of checks in preparation for peer review. Once your manuscript has passed the checks it will be sent out for review. To provide the metadata for your submission, please Login to Editorial Manager (https://www.editorialmanager.com/pbiology) within two working days, i.e. by Dec 22 2021 11:59PM.

Please note that due to the upcoming holiday period, there may be some delays in recruiting the previous reviewers during the re-review process. Thank you in advance for your patience with this and we wish you happy holidays. 

If your manuscript has been previously reviewed at another journal, PLOS Biology is willing to work with those reviews in order to avoid re-starting the process. Submission of the previous reviews is entirely optional and our ability to use them effectively will depend on the willingness of the previous journal to confirm the content of the reports and share the reviewer identities. Please note that we reserve the right to invite additional reviewers if we consider that additional/independent reviewers are needed, although we aim to avoid this as far as possible. In our experience, working with previous reviews does save time. 

If you would like to send previous reviewer reports to us, please email me at rhodge@plos.org to let me know, including the name of the previous journal and the manuscript ID the study was given, as well as attaching a point-by-point response to reviewers that details how you have or plan to address the reviewers' concerns. 

Given the disruptions resulting from the ongoing COVID-19 pandemic, please expect some delays in the editorial process. We apologise in advance for any inconvenience caused and will do our best to minimize impact as far as possible.

Kind regards,

Richard

Richard Hodge, PhD

Associate Editor, PLOS Biology

rhodge@plos.org

PLOS

---

## [Decision Letter · Decision Letter 1]

9 Feb 2022

Dear Dr Galli,

Thank you for submitting a a revised version of your manuscript "Endomitosis controls tissue-specific gene expression during development" for consideration as a Research Article at PLOS Biology. Please accept my sincere apologies for the delays that you have experienced during the peer review process. Your manuscript has been evaluated by the PLOS Biology editors, an Academic Editor with relevant expertise, and by the three original reviewers from Review Commons.

The reviews are attached below. As you can see, the reviewers are generally positive about the revised manuscript. However, Reviewer #2 raises concerns with the auxin-inducible KNL-1 depletion line used to block binucleation in the GFP reporter assays. The reviewer suggests that binucleation is more directly blocked using a CDK-1 depletion line instead. In addition, Reviewer #3 asks for some additional discussions and contextualization of the model presented in the manuscript text.

After discussions with the academic editor, we strongly encourage you to include the auxin-inducible CDK-1 depletion experiment in the revised manuscript, as this would strengthen the accuracy of the conclusions presented. However, we could consider a discussion of the limitations of using the KNL-1 depletion line in the manuscript text if the experiment will take a long time to set up and complete. 

In light of the reviews, we will not be able to accept the current version of the manuscript, but we would welcome re-submission of a much-revised version that takes into account the reviewers' comments. We cannot make any decision about publication until we have seen the revised manuscript and your response to the reviewers' comments. Your revised manuscript is also likely to be sent for further evaluation by the reviewers.

We expect to receive your revised manuscript within 3 months. Please email us (plosbiology@plos.org) if you have any questions or concerns, or would like to request an extension. At this stage, your manuscript remains formally under active consideration at our journal; please notify us by email if you do not intend to submit a revision so that we may end consideration of the manuscript at PLOS Biology.

**IMPORTANT - SUBMITTING YOUR REVISION**

*Re-submission Checklist*

*Published Peer Review*

*PLOS Data Policy*

*Blot and Gel Data Policy*

Sincerely,

Richard

Richard Hodge, PhD

Associate Editor, PLOS Biology

rhodge@plos.org

REVIEWS:

Reviewer #1: Authors have addressed all my original concerns and should be commended on their rigorous study to elucidate the physiology of binucleation in the C.elegans intestinal cells. The manuscript reads well and clearly demonstrates the transcriptional adaption of binucleation and its impact of worm fecundity. This is the first study to date to clearly show in vivo that a cell with multiple nuclei enhances its gene expression compared to a mononucleated cell with equal ploidy.

Reviewer #2: In our original review for van Rijnberk et al, we expressed enthusiasm for the development of methods to perturb endomitosis in C. elegans and the potential importance of binucleation in driving the rapid expression of vitellogenins in intestine cells. In this new version, the authors have addressed the major concerns regarding the link between vitellogenin expression and the different phenotypes observed. While the observed defects in brood size and gene expression are relatively mild, they nonetheless appear to be reproducible.

Interestingly, the authors' data suggest that binucleation is critical for driving rapid gene expression at the larva to adult transition (Fig. 6). To test this idea, the authors developed an assay consisting of a heat-shock promoter driving the expression of GFP. While in controls, GFP is rapidly expressed upon heat-shock, in cells with no endomitosis GFP expression is delayed. These experiments were performed using a line that expressed auxin-inducible KNL-1 degradation to prevent binucleation, but whether the effects observed are a consequence of loss of binucleation or an indirect effect of loss of KNL-1 in chromosome segregation is currently unclear. Considering the importance of this experiment to the overall conclusions of this study, it would be useful to test this idea using a different perturbation that blocks binucleation, such as auxin-inducible CDK-1 depletion. Alternatively, the authors should explain the reason why they chose to use the AID::KNL-1 line for these experiments and not the AID::CDK-1 one.

Minor comment: There is an error in the labelling of Fig. 7a.

Reviewer #3: The authors of the manuscript number PBIOLOGY-D-21-03267R1 entitled "Endomitosis controls tissue-specific gene expression during development" have addressed adequately all my previous comments from their first submission. I have a few of comments/questions. 

1. The authors propose that CEH-60 and UNC-62 are limiting VIT-2 in mononucleated intestinal cells. Are they expressed differentially in mononucleated cells or are they suggesting they are translated at lower level in these cells?

2. Other genes downregulated in the intestine include a ribosomal protein (rpl-11.2), and a transcription factor (ets-4) predicted to enable DNA-binding transcription factor activity, RNA polymerase-specific (from WormBase). Lower levels of these two transcripts would be expected to downregulate translation globally and/or RNA polymerase II-specific transcription. How does this possibility reconcile with your findings? Do the authors think it should be addressed in the discussion? 

Importantly, the authors do not mention in the discussion or elsewhere that ets-4 transcription factor positively regulates vit-2, vit-3, vit-4 and vit-5, as well as ceh-60. Nor that mutations in ets-4 result in reduced gene expression levels.

3. This is just a comment. It is a bit surprising that overexpression of CEH-60 or UNC-62 completely rescue the phenotypes of animals with mononucleated intestines, given other downregulated genes (spp-5, cpl-1) also affect brood size and/or affect fat content/processing. Should this be a concern or "explained" why might be so?

---

## [Editor Report · Decision Letter 2]

28 Feb 2022

Dear Dr Galli,

Thank you for submitting your revised Research Article entitled "Endomitosis controls tissue-specific gene expression during development" for publication in PLOS Biology. Please accept my apologies for the delay in getting back to you. I have now obtained advice from the Academic Editor handling your submission, who is satisfied that the reviewer concerns have been fully addressed.

Based on this, we will probably accept this manuscript for publication, provided you address the following data and other policy-related requests that I have provided below (A-D):

(A) You may be aware of the PLOS Data Policy, which requires that all data be made available without restriction: http://journals.plos.org/plosbiology/s/data-availability. For more information, please also see this editorial: http://dx.doi.org/10.1371/journal.pbio.1001797

Regardless of the method selected, please ensure that you provide the individual numerical values that underlie the summary data displayed in the following figures, as they are essential for readers to assess your analysis and to reproduce it.

Figure 1E, 1G, 1H, 2A-B, 2D-H, 2K-M, 3A, 4A, 4C-G, 5B-G, 6A, 6D-E, 7A-D, S1B, S1D-E, S2B-D, S3A-F, S4A-G, S5B-E, S6A-B, S7, S8

B) Please also ensure that each of the relevant figure legends in your manuscript include information on *WHERE THE UNDERLYING DATA CAN BE FOUND*, and ensure your supplemental data file/s has a legend.

(C) Thank you for depositing the RNA-Sequencing data in the GEO database and for providing the accession number. However, we note that GSE169330 is currently private and is scheduled to be released in September 2022. You should make this data publicly available now for us to be able to process your manuscript for Production.

(D) Please ensure that your Data Statement in the submission system accurately describes where your data can be found and is in final format, as it will be published as written there. 

We expect to receive your revised manuscript within two weeks. 

*Published Peer Review History*

*Early Version*

Sincerely,

Richard

Richard Hodge, PhD

Associate Editor, PLOS Biology

rhodge@plos.org

PLOS

---

## [Editor Report · Decision Letter 3]

9 Mar 2022

Dear Dr Galli,

On behalf of my colleagues and the Academic Editor, Tom Misteli, I am pleased to say that we can in principle accept your Research Article "Endomitosis controls tissue-specific gene expression during development" for publication in PLOS Biology, provided you address any remaining formatting and reporting issues. These will be detailed in an email that will follow this letter and that you will usually receive within 2-3 business days, during which time no action is required from you. Please note that we will not be able to formally accept your manuscript and schedule it for publication until you have any requested changes.

PRESS

Sincerely, 

Richard

Richard Hodge, PhD

Associate Editor, PLOS Biology

rhodge@plos.org

PLOS
